

# Spatial distribution and seasonal variability in atmospheric ammonia measured from ground-based FTIR observations at Hefei, China

Wei Wang[1], Cheng Liu[2,3,1,4,5*], Lieven Clarisse[6], Martin Van Damme[6], Pierre-François Coheur[6], Yu Xie[7], Changgong Shan[1], Qihou Hu[1], Huifang Zhang[1], Youwen Sun[1], Hao Yin[1], Nicholas Jones[8,9*]

[1]Key Laboratory of Environmental Optics and Technology, Anhui Institute of Optics and Fine Mechanics, Chinese Academy of Sciences, Hefei, 230031, China

[2]Department of Precision Machinery and Precision Instrumentation, University of Science and Technology of China, 230026 Hefei, China

[3]Center for Excellence in Regional Atmospheric Environment, Institute of Urban Environment, Chinese Academy of Sciences, Xiamen, 361021, China

[4]Key Laboratory of Precision Scientific Instrumentation of Anhui Higher Education Istitutes, University of Science and Technology of China, Hefei, 230026, China

[5]Anhui Province Key Laboratory of Polar Environment and Global Change, University of Science and Technology of China, Hefei, 230026, China

[6]Université libre de Bruxelles (ULB), Atmospheric Spectroscopy, Service de Chimie Quantique et Photophysique, 1050 Brussels, Belgium

[7]Department of Automation, Hefei University, Hefei 230601, Anhui, China

[8]School of Earth, Atmospheric and Life Sciences, University of Wollongong, Northfields Ave, Wollongong, NSW, 2522, Australia

[9]School of Physics, University of Wollongong, Northfields Ave, Wollongong, NSW, 2522, Australia

*Correspondence to*: Cheng Liu (chliu81@ustc.edu.cn)
             Nicholas Jones (njones@uow.edu.au)

**Abstract**: Atmospheric ammonia ($NH_3$) plays an important role in the formation of fine particulate matter, leading to severe environmental degradation and human health issues. In this work, ground-based FTIR observations are used to obtain the total columns and vertical profiles of atmospheric $NH_3$ at a measurement site in Hefei, China, from December 2016 to November 2018. The spatial distribution and temporal variation, seasonal trend, emission sources and potential sources of $NH_3$ are analyzed. The time series of ammonia columns show that FTIR observations captured the seasonal cycle of $NH_3$ over the two years of measurement, with a 22.14 % yr$^{-1}$ annual increase rate over the Hefei site. We used IASI satellite data to compare with the FTIR data, and the correlation coefficients (R) between the two datasets are 0.86 and 0.78 for IASI-A and IASI-B, respectively. The results demonstrate the IASI data are in broad agreement with our FTIR data. To examine the contribution of traffic to $NH_3$ columns, we analyze the relationship of $NH_3$ columns with CO surface concentrations. $NH_3$ columns show high correlation

(R=0.77) with CO concentrations in summer, indicating that the elevated $NH_3$ columns are partly caused by urban emissions from vehicles. Further, high correlation of $NH_3$ columns with air temperature is obvious from their diurnal variation during the observation period. In addition, the clear correlation

between $NH_3$ columns and air temperature in spring and autumn over Hefei, suggests that agriculture was indeed the main source of ammonia in spring and autumn. Furthermore, the back trajectories of air masses calculated by the HYSPLIT model confirmed that agriculture was the dominant source of ammonia in spring, autumn and winter, while urban anthropogenic emissions contributed to the high level of $NH_3$ in summer over the Hefei site. The potential source areas influencing the $NH_3$ columns were

distributed in the local area of Hefei, the northern part of Anhui province, as well as Shangdong, Jiangsu and Henan provinces. This study helps to identify the emission sources and potential sources that contribute to $NH_3$ columns over Hefei, a highly populated and polluted area. This is the first time that ground-based FTIR remote sensing of $NH_3$ columns and comparison with satellite data are reported in China.

**1 Introduction**

Atmospheric ammonia ($NH_3$) plays a significant role in the formation of fine particulate matter ($PM_{2.5}$), as ammonia reacts rapidly with nitric acid and sulfuric acid to form ammonium salts (Behera et al., 2013; Meng et al., 2018). Ammonium salts constitute a large proportion of fine particulate matter, which has an adverse impact on air quality, climate change and human health (WHO, 2013). Moreover, ammonium

salts have a longer atmospheric lifetime (several days) than that of gaseous $NH_3$ (hours to days), so that they can be subject to long-range transport away from $NH_3$ sources. Globally, the main sources of atmospheric ammonia are related to agricultural activities, including farming and animal husbandry (Sutton et al., 2013). Atmospheric ammonia also originates from other sources, such as biomass burning, vehicle exhaust, natural vegetation and wild animals. Recently, industrial emissions have also been

identified as important point source emitters (Van Damme et al., 2018). Major sinks of atmospheric ammonia are dry deposition, wet removal by precipitation, and conversion to particulate ammonium salts via reaction with acids (Baek et al., 2005; Liu et al., 2011).

Although ammonia is a major player in various environmental and health issues, the ammonia budget and its contribution of specific sources to emissions still remain uncertain on regional scales. This is



mainly caused by lack of representative measurements of atmospheric $NH_3$. In the last two decades, there

have been significant efforts to measure atmospheric $NH_3$ around the world. In situ measurements based

on passive samplers or dedicated denuders are usually performed with a time resolution of days to weeks

(Erisman et al., 2001; Perrino et al., 2002; Adon et al., 2010; Cisneros et al., 2010; Pinder et al.,2011;

Day et al., 2012; Heald et al., 2012; Zbieranowski et al., 2012; Makkonen et al., 2012; Benedict et al.,

2013; Li et al., 2014; Li et al., 2017). However, atmospheric ammonia at ambient levels is difficult to

measure in situ, due to its reactive and sticky nature, the rapid gas-to-particle conversion in the

atmosphere, and the strong spatial and temporal variations of concentration. Only a few sites use

spectroscopic measurement techniques, such as quantum cascade laser absorption spectroscopy (QCLS),

differential optical absorption spectroscopy (DOAS), cavity-ring down spectroscopy (CRDS), open path

FTIR and photoacoustic spectroscopy to provide high temporal resolution data (Pogány et al., 2009; Von

Bobrutzki et al., 2010; Volten et al., 2012; Miller et al., 2014; Sun et al., 2014; Dammers et al., 2015;

Sintermann et al., 2016; Berkhout et al., 2017; Benedict et al., 2017; Phillips et al., 2019). Compared to

ground-based in situ observations, measurements of the $NH_3$ vertical profiles from in situ observations

are sparser. Recently, airborne and tower measurements to obtain vertical profiles of $NH_3$ in the free

troposphere have appeared (Yokelson et al., 2003; Nowak et al., 2012; Leen et al., 2013; Schiferl et al.,

2014; Battye et al., 2016 ; Dammers et al., 2017a; Li et al., 2017).

During the last decade, satellite have shown improving abilities to monitor global and regional

distributions of $NH_3$. $NH_3$ column densities have been obtained by the Tropospheric Emission

Spectrometer (TES) instrument on the NASA EOS Aura satellite (Beer et al., 2008; Shephard et al., 2011;

Pinder et al., 2011), the Infrared Atmospheric Sounding Interferometer (IASI) instrument on the Metop

satellite (Clarisse et al., 2009; Clarisse et al., 2010; Van Damme et al., 2014a), the Cross-track Infrared

Sounder (CrIS) instrument on the Suomi National Polar-orbiting Partnership (NPP) satellite (Shephard

and Cady-Pereira, 2015), and the Atmospheric InfraRed Sounder (AIRS) instrument on the NASA EOS

Aqua satellite (Warner et al., 2016). $NH_3$ concentrations in the upper troposphere were detected by

Michelson Interferometer for Passive Atmospheric Sounding (MIPAS) on board the Envisat satellite by

analyzing the infrared limb-emission spectra (Höpfner, et al., 2016). Satellite observations have been

applied in air quality monitoring, quantification of source emissions, trend analysis, and model evaluation

(e.g., Van Damme et al., 2018; Clarisse et al., 2019; Dammers, et al., 2019; Lachatre, et al., 2019).

Nonetheless, satellite data are limited by large uncertainties due to atmospheric conditions (mainly



thermal contrast, and cloud coverage), and usually less accurate than ground-based measurements. Satellite data need to be validated by high-precision and high-accuracy data obtained independently by ground-based instruments. Ground-based FTIR observations have been commonly used to validate satellite data products, such as carbon dioxide ($CO_2$), methane ($CH_4$), carbon monoxide (CO), and nitrous oxide ($N_2O$) (Dils, et al., 2006; Morino et al., 2011; Reuter et al., 2011; Schneising et al., 2012). More

recently, FTIR measurements have been shown to also provide total column and vertical profiles of ammonia at a high temporal resolution, and are now also used for validation of satellite $NH_3$ observations (Dammers, et al., 2015; Dammers, et al., 2016; Dammers, et al., 2017b). Moreover, FTIR $NH_3$ data have been used to measure ammonia emissions from biomass burning and demonstrate long-range transport of $NH_3$ (Paton-Walsh et al., 2005; Lutsch et al., 2016; Lutsch et al., 2019).

High levels of ambient ammonia has become one of the most prominent air pollution problems in recent years and given rise to growing concerns in China. The ammonia emission inventory developed by the Peking University reveals that the national $NH_3$ emissions increased by 64.6%, from 5.9 to 9.7 Tg from 1980 to 2012 (Kang et al., 2016). According to the EDGAR emission inventory, the total $NH_3$ emissions grew by 357.8%, from 3.06 to 14.01 Tg from 1970 to 2010 in China (EDGAR, 2016). The 14-year AIRS

satellite data record indicates the significant increasing trends of $NH_3$ concentration for this country, with the rate of 2.27 % $yr^{-1}$ from 2002 to 2016 (Warner et al., 2017). The two major contributors of ammonia emission in China, livestock manure and synthetic fertilizer application contributed to 80–90% of the total emissions from 1980 to 2012 (Kang et al., 2016). $NH_3$ emissions are predicted to continue to increase in the next few years, owing to ongoing increases in fertilizer application and intensive livestock

farms. From the spatial distribution of $NH_3$ emissions derived from the inventory, it is seen that virtually all high levels of $NH_3$ concentrations are above the agricultural regions of China, such as the North China Plain (Hebei, Shandong, Henan, Jiangsu, Anhui) and Sichuan provinces (Huang et al., 2012; Kang et al., 2016). This is in agreement with the spatial pattern of $NH_3$ distributions observed by satellite (Van Damme et al., 2014a; Warner et al., 2016; Warner et al., 2017). Emission inventories and satellite

observations of $NH_3$ need validation from ground-based remote sensing in China. Furthermore, ground-based observations of $NH_3$ have been sparse throughout China (Liu et al., 2011; Xu et al., 2015).

Despite the importance of $NH_3$ in the formation of particulate ammonium, $NH_3$ emission in China has not been routinely monitored in contrast to sulfur dioxide ($SO_2$), nitrogen oxides ($NO_x$), CO, and fine particles, due to the lack of specific regulatory requirements for its measurement. The National program



on energy saving and emission reduction published by the Chinese State Council aims to reduce $SO_2$ emissions in the eleventh Five-Year Plan (2005–2010) (http://www.gov.cn/zhengce/content/2008-03/28/content_5007.htm), while the twelfth Five-Year Plan (2011–2015) aims to reduce, in addition to emissions of $SO_2$, those of ammonia and nitrogen oxides, starting to control the emissions of ammonia from agriculture (http://www.gov.cn/zhengce/content/2011-09/07/content_1384.htm). Moreover, the

Action Plan of Air Pollution Prevention and Control issued by the Chinese State Council in 2013 requires that the concentration of the inhalable particles reduces more than 10 % between 2012 and 2017 in cities at the prefectural level and above, and the fine particle concentration reduces 25 %, 20 % and 15 % in the Beijing-Tianjin-Hebei region, Yangtze River Delta, and Pearl River Delta, respectively (http://www.gov.cn/zhengce/content/2013-09/13/content_4561.htm). Reduction of $NH_3$ emissions is

considered an effective way to lower $PM_{2.5}$ pollution (Erisman et al., 2004; Wang et al., 2015; Wu et al., 2016).

Hefei, located in eastern China, is a highly populated and polluted region, with intensive agricultural production, heavy traffic and transportation. Hefei has suffered severe haze and poor visibility in recent years (Hong, et al., 2018; Tan et al., 2019). Concentrations of $PM_{2.5}$ often exceed the Ambient Air Quality

Standard in autumn and winter. Although many studies are aiming to understand particulate matter pollution in China, little is known about the role and contribution of $NH_3$ in fine particulate formation on a regional or local scale. In this study, we present and analyze temporal and spatial distribution, seasonal trends, emission sources and potential source areas retrieved from two years of ground-based FTIR measurements of $NH_3$.

This paper is organized as follows. Materials and data are described in Section 2, in particular, the measurement site and instrumentation, the retrieval methods of $NH_3$, and IASI satellite data are introduced. Results and discussion are presented in Section 3. The vertical distribution of $NH_3$ and characteristics are shown in Section 3.1. Time series, seasonal trends and annual variability are analyzed in Section 3.2. Comparisons of ground-based measurements with satellite data for $NH_3$ are made in

Section 3.3. The relation of $NH_3$ with surface CO concentrations is discussed in Section 3.4. Then, the potential sources that contribute to $NH_3$ columns over Hefei are identified based on analysis of the relationship of $NH_3$ with meteorological parameters (Section 3.5). Conclusions are presented in Section 4.



## 2 Materials and data

### 2.1 Site description and instrumentation

The Hefei site (31°54′ N, 117°10′ E, 29 m above sea level), part of the Anhui Institute of Optics and Fine Mechanics, is operated by Key Laboratory of Environmental Optics and Technology, Chinese Academy of Sciences. It is located in the north-western rural area of Hefei city in eastern China (Fig. 1), adjacent to the Dongpu lake in a flat terrain. The Hefei urban area, about 10 km south-east of the site, is densely populated with about 7.7 million people. The site is surrounded by wetlands or cultivated lands in other directions.

A Bruker IFS 125HR FTIR spectrometer and a solar tracker are combined to routinely measure trace gases since January 2014. The FTIR spectrometer and the solar tracker are detailed in Wang et al. (2017). The spectrometer uses a liquid-nitrogen-cooled MCT/InSb detector in combination with a KBr beamsplitter and a suit of optical filters to record mid-infrared (MIR) solar absorption spectra (700-4000 $cm^{-1}$) since July 2015. The solar spectra in the 700-1350 $cm^{-1}$ filter region, obtained with the MCT detector at a spectral resolution of 0.005 $cm^{-1}$ are used to retrieve $NH_3$.

Additionally, a weather station (ZENO, Coastal Environmental Systems, USA) mounted near the solar tracker on the roof recorded meteorological parameters, such as surface pressure, air temperature, relative humidity, wind speed, wind direction, solar radiation, rain, snow and leaf wetness since September 2015. At the same time, the indoor pressure, temperature and relative humidity are logged continuously.

### 2.2 Retrieval methods

Two spectral micro-windows were chosen to retrieve atmospheric ammonia, similar to the $NH_3$ retrieval strategies in Dammers et al. (2015). The first micro-window (MW1) covers the spectral range of 929.4–931.4 $cm^{-1}$. The interfering species in MW1 are $H_2O$, $O_3$, $CO_2$, and two isotopologues of $CO_2$ ($^{13}CO_2$, and $C^{16}O^{18}O$). The second micro-window (MW2) spans the spectral range of 962.1–970.0 $cm^{-1}$ and is characterized by the same interfering species. The retrieval is performed using the SFIT4_0.9.4.4 algorithm (an updated version of SFIT2, Rinsland et al, 1998), which is based on the optimal estimation method to retrieve vertical profile of concentration and total columns of $NH_3$. A priori information, including gas vertical profiles and covariance matrices are used to constrain the retrieval. A priori profiles of $NH_3$ and interfering gases are taken from the Whole Atmosphere Community Climate Model (WACCM, v.6_120_99) in combination with initial measurement values. The a priori covariance matrix

for ammonia was constructed to be diagonal, with standard deviations of 100% for all layers. The temperature and pressure profiles for the meteorological parameters are taken from the National Centers

for Environmental Prediction (NCEP) analysis for each day. The profiles were separated into 48 discrete layers for the forward model calculations, from the surface up to 120 km. The HITRAN 2012 spectral database is used for the spectroscopic line parameters. Figure 2 shows a typical spectral fit of $NH_3$ in the spectral windows centered at 930.4 and 966.05 $cm^{-1}$, respectively. The measured spectrum is shown in blue, the fitted spectrum in red and the residual in black. The RMS value of the residuals is used to judge

the quality of the fits for each of the retrievals. The RMS of the residuals is about 0.498 % and 0.505% in the two spectral windows, respectively.

### 2.3 IASI data

IASI are Fourier transform spectrometers onboard the platforms Metop-A and Metop-B, launched in October 2006 and September 2012, respectively. The platforms Metop circle in a polar sun-synchronous

orbit around the earth. IASI operates in nadir mode (vertically downward) to measure the infrared radiation emitted from the surface of the Earth and its atmosphere. It provides global coverage twice a day by scanning along a swath of 2,200 km off-nadir. The mean local solar overpass times are 9:30 am and 9:30 pm at the equator. IASI measures the radiances in the thermal infrared spectral range of 645–2,760 $cm^{-1}$, with an apodized spectral resolution of 0.5 $cm^{-1}$. Clerbaux et al. (2009) provides a detailed

description of the IASI instrument.

The IASI $NH_3$ data used here are part of the ANNI-$NH_3$-v3R retrieval product (Van Damme et al., 2014a; Whitburn et al., 2016; Van Damme et al., 2017; Franco et al., 2018). A few comparison studies have been performed to validate the IASI-$NH_3$ data product using independent ground-based or airborne measurements (Van Damme et al., 2015a; Dammers et al., 2016). These validations indicate a general

good agreement, but also the possible presence of small biases. The IASI data products have been used to estimate $NH_3$ emissions from agricultural sources or biomass burning, to evaluate model simulations, and to identify small emission sources (Van Damme et al., 2014b; Whitburn et al., 2015; Fortems-Cheiney, et al., 2016; Schiferl et al., 2016; Li et al., 2017; Van Damme et al., 2018). In our study, only the IASI data collected from the morning orbit are considered, as the sensitivity of thermal nadir measurements

near the surface is higher at this time, owing to a larger thermal contrast in most places.



## 3 Results and discussion

### 3.1 Characteristics of vertical distribution of NH$_3$

#### 3.1.1 Error analysis of NH$_3$ retrieval

An error analysis was performed on the basis of the error estimation method described in Rodgers (1990). The error calculation is based on attributing uncertainties to all parameters used in the profile retrieval. The influence of such parameters as the temperature profile, solar zenith angle (SZA), spectroscopic line parameters and interfering species has been studied. The error budget can be divided into three contributions: the model parameter error due to the inaccurately-described forward model parameters,

the measurement error due to the measurement noise, and the smoothing error due to the low vertical resolution of the retrieval. Table 1 lists the uncertainties of the parameters assumed in the retrievals. The results of error analysis for a typical NH$_3$ retrieval are summarized in Table 2. The total errors are about 11.42 % based on the combination of random and systematic errors. The random error is mainly due to temperature uncertainty and measurement noise, with an error of 2.56 %. As for the systematic

error, it amounts to an error of 11.13 %, dominated by uncertainties in spectroscopic line parameters, with small contributions from uncertainties in temperature, SZA, and phase. It is clear that uncertainties of the line intensity parameter for the ammonia absorption lines are the main error sources for the NH$_3$ retrieval.

#### 3.1.2 Vertical distribution of NH$_3$

Figure 3 displays the a priori profile and the seasonal averaged vertical profiles of NH$_3$ at the Hefei site. The seasons are expressed as three-month periods, from December to February (winter), from March to May (spring), from June to August (summer), and from September to November (autumn). The retrieved profiles show that the concentration of NH$_3$ peaked near the surface in all four seasons. Moreover, summer presents the maximum of ammonia levels, and the values observed in autumn are comparable

to those in spring, while the minimum of NH$_3$ appeared in winter. The seasonal averaged surface level of NH$_3$ decreased from 10.82 ppb in summer to 2.92 ppb in winter during 2017 and 2018, and the corresponding values are about 5.48 and 6.04 ppb in spring and autumn, respectively.

The layer averaging kernels and total column averaging kernel for a typical profile retrieval of NH$_3$ on the 48-layer height grid are shown in Figure 4. It is evident that the retrieval is most sensitive to the



troposphere, where the concentration of ammonia peaks. The degrees of freedom for signal (DOFS) value

is 1.10 given by this measurement (30 August 2018, 10:33 Local time; solar zenith angle: 32.34°; $NH_3$

total column: $1.64\times10^{16}$ molec $cm^{-2}$), which is a typical value obtained from ammonia retrievals at the

Hefei site. The DOFS of 1.10 indicates that the averaging kernels are not vertically resolved and there is

almost no vertical information available for multiple layers. The DOFs provided by $NH_3$ retrievals in

Hefei site are similar to the results from the FTIR measurement at other sites (Dammers et al., 2015;

Dammers et al., 2017b).

### 3.2 Time series and seasonal trend of $NH_3$

The time series of the ammonia column observed by the FTIR from December 2016 to November 2018

at the Hefei site are plotted in Figure 5. The data are not continuous, with gaps due to adverse weather

conditions. Many spectra ranging from 700 to 1350 $cm^{-1}$ are saturated in summer (due to high humidity),

causing the retrieved $NH_3$ data to be sparsely sampled relative to those in other seasons. The seasonal

and inter-annual variations of ammonia are clearly identifiable. A combination of sine and cosine

trigonometric functions was used to fit the seasonal variation of $NH_3$ columns, expressed in Eq. (1)

(Keeling et al., 1976; Thoning et al., 1989), where $X$ represents the individual $NH_3$ columns, $t$ is the

elapsed time in years, $A_0$ denotes the initial state of $NH_3$ column, $A_1$ is the slope of the linear part, and

$A_2$–$A_5$ are the fitting coefficients describing the seasonal cycle. The parameters in Eq. (1) are detailed in

the study of Bie et al. (2018).

$$X(t) = A_0 + A_1 t + A_2 \sin 2\pi t + A_3 \cos 2\pi t + A_4 \sin 4\pi t + A_5 \cos 4\pi t \quad (1)$$

The seasonal amplitude of $NH_3$ column over the Hefei site is comparable for the two years, with value

of $5.33\times10^{16}$ and $5.42\times10^{16}$ molec $cm^{-2}$, respectively. The maximum and minimum $NH_3$ appear in

summer and winter, respectively. The annual mean $NH_3$ column is $1.31\times10^{16}$ and $1.60\times10^{16}$ molec $cm^{-2}$,

respectively, with an increase rate of about 22.14 %.

The summer maximum and winter minimum of $NH_3$ over the two years indicate that agricultural

practices maybe the main source of $NH_3$ over the Hefei site. In the recent study by Shephard et al. (2011),

the representative volume mixing ratio (RVMR) of $NH_3$ observed by TES satellite over southeast China

(22°N to 42°N, 99°E to 121°E) exhibited a distinct seasonal cycle, with peak concentration in summer.

Van Damme et al. (2015a) found that $NH_3$ columns observed by IASI and concentrations from the surface

measurements during 2008 through 2014 both peaked in summer at the Shangzhuang site in the northwest



of Beijing, China, which is surrounded by agriculture. In Van Damme et al. (2015b), six years of IASI

measurements from 2008 to 2013 reported the seasonal variability of ammonia columns in southeastern

China (22°N to 42°N, 98°E to 122°E), and the summer peaks were consistent with the in situ

measurement data. In Warner et al. (2016), measurements made by AIRS from 2002 through 2015 show

that, the $NH_3$ seasonal pattern over cropland areas in east-central China are similar to those over cropland

areas in the northern hemisphere, with the highest columns in summer and spring. It follows therefore

that seasonal variation of $NH_3$ columns in the Hefei area accords with that in other areas in China, with

the main emission source being agriculture.

The recent study of Liu et al. (2017) derived that $NH_3$ column increased at a rate of 2.37 % yr$^{-1}$ over

China from IASI observations during warm months of 2008 to 2014. In Warner et al. (2017), AIRS

observations from 2002 to 2016 over east central China, one of the world's major agricultural regions,

revealed a 2.27 % yr$^{-1}$ increasing trend of atmospheric $NH_3$ concentrations. This annual increase rate of

$NH_3$ agrees with 2.7 % yr$^{-1}$ increase in the use of fertilizer in China. The annual increasing rate of

ammonia columns in Hefei estimated by our two-year FTIR measurements (22.14 % yr$^{-1}$) is much larger

than the reported value by satellite observations over China. This is likely due to the different sampling

years. The increasing trend of $NH_3$ in Hefei is likely caused by either an increased fertilizer use, or

increasing air temperature, or decreased sulfur emissions due to strict $SO_2$ control measures.

**3.3 Comparison with satellite data**

Here we present a comparison with the IASI satellite measurements. The FTIR dataset is suitable for

comparison with satellite data given high concentrations observed and the flat geography surrounding

the Hefei station. For comparison with ground-based FTIR measurements, IASI Level 2 product data

within 0.5° latitude/longitude radius of Hefei station were considered. We set the collocation time to 90

minutes. We remove the data with negative IASI-$NH_3$ columns due to large retrieval error. Table 3 details

the data filtering criteria, which follows the criteria adopted in Dammers et al. (2016).

In order to compare two measurements from different remote-sensing instruments directly, their different

vertical sensitivity and a priori profiles should be accounted for (Rodgers and Connor, 2003). Since the

IASI-$NH_3$ retrieval does not provide averaging kernels and vertical profiles (Van Damme, et al 2014a),

this method for comparison is not applied. Here we therefore compare the IASI satellite and FTIR data

directly, without considering the effect of different a priori profiles and averaging kernels.





Figure 6 depicts the direct comparison of our data with respect to the co-located IASI V3R data. The IASI data are averaged if multiple satellite overpasses match one single FTIR observation. Although there are a few data matching these coincidence criteria, it is found that the IASI data are in broad agreement with the FTIR data. The mean relative difference between IASI and ground-based FTIR columns are computed (satellite minus FTIR, divided by FTIR), and the standard deviation of the relative differences are also calculated. The Relative differences larger than 100% were considered as outliers from the data. There are 230 and 264 pairs of matched data for columns of $NH_3$ for IASI-A and IASI-B satellite data, giving mean relative difference of 4.51% and 0.33%, with standard deviation of 44.44% and 41.00%, respectively. The correlation coefficients R are 0.86 and 0.78, respectively. The scatter graphs of the retrieval results of FTIR and IASI in Figure 6 (a) and (b) show a good linear relationship. Additionally, the distributions of the relative difference of the two dataset show that, the relative bias mainly range from -60% to 80% for IASI A data, from -60% to 60% for IASI B data, and the bias from -20% to 0% as well as -40% to -20% has the highest frequency in both respective bins (Figure 6 (c) and (d)).

Dammers et al. (2016) first validated the IASI $NH_3$ data product using ground-based FTIR observations from nine NDACC stations, and showed that the mean relative difference between the satellite $NH_3$ total columns and FTIR data were -32.4 ± (56.3) %, with a correlation coefficient of 0.8. Dammers et al. (2017b) compared CrIS $NH_3$ column and profile data with FTIR measurements from seven co-located NDACC stations. The correlation coefficient (R) between the CrIS $NH_3$ total columns and FTIR data was 0.77, and the relative difference is 0-5 % with a standard deviation of 25–50 % for comparison of high levels of $NH_3$. The average relative difference between the CrIS and FTIR profile was in the range of 20 to 40 %. So the relative differences between IASI total columns and our FTIR data and standard deviations of the differences are within the range of comparison results from other NDACC site data, and the correlation coefficients are comparable to that of other comparison results.

**3.4 Relationship of $NH_3$ with surface CO**

The number of cars in mid-2017 reached more than 1.5 million in Hefei according to the report of the Hefei Traffic Management Bureau, and vehicle exhaust has become an important source of urban air pollution. The tunnel studies carried out in China and other countries indicate that motor-vehicle exhausts constitute an important source of $NH_3$ in urban areas (Perrino et al., 2002; Ianniello et al., 2010; Sun et





al., 2014; Chang et al., 2016). The correlation between $NH_3$ and CO concentrations reported in these studies suggests a common source for urban $NH_3$ and CO, as CO is a traffic emitted primary non-reactive pollutant. To examine the contribution of traffic to $NH_3$ columns, we analyze the relationship of $NH_3$

columns with CO surface concentrations. The Dongpu Reservoir air quality monitoring station (31.91°N, 117.16°E) is very close to our site, part of a National Ambient Air Quality Monitoring Network, which monitors and routinely publishes the concentrations of main gaseous pollutants, including CO, $NO_2$, $PM_{2.5}$, $PM_{10}$, $SO_2$, $O_3$ and Air Quality Index (AQI) etc. The air quality data are daily averaged. There is no routine measurement of surface $NH_3$ concentrations at the monitoring site. The relationship between

$NH_3$ columns observed with surface concentrations of CO, $NO_2$, $SO_2$, $PM_{2.5}$ and $PM_{10}$ over different seasons is analyzed, respectively. The results show that there exists no correlation between $NH_3$ columns and concentrations of $NO_2$, $SO_2$, $PM_{10}$ and the value of AQI over the four seasons (not shown).

However, $NH_3$ columns show high correlation with CO concentrations in summer, as displayed in Figure 7(a). The correlation coefficient (R) is 0.77, although the summer data are sparse. Because $NH_3$ is mainly

distributed in the boundary layer, $NH_3$ columns represent the surface concentrations of $NH_3$ to some extent. The close link between $NH_3$ columns and CO concentrations indicates that $NH_3$ has common sources with CO in summer over the Hefei site. Atmospheric CO is regarded as a primary pollutant mainly emitted from vehicles in urban areas, and there is no significant biomass burning source around the Hefei site, thus the elevated $NH_3$ columns are likely partly caused by urban emissions from vehicles.

Meanwhile, $NH_3$ columns show weak correlation (R=0.47) with $PM_{2.5}$ concentrations (Fig. 7(b)), meaning that $NH_3$ contributed to the formation of fine particulates significantly in summer.

### 3.5 Identification of potential source of $NH_3$

The variability of $NH_3$ columns is strongly affected by emission strengths from agricultural sources, and meteorological and atmospheric conditions, such as air temperature, wind speed, wind direction and

relative humidity. To assess the impact of meteorological parameters on the variation of $NH_3$ columns, we analyze the relationship between $NH_3$ columns with these meteorological and atmospheric conditions. High correlation of $NH_3$ columns with air temperature is obvious from their diurnal variation during the observation period, as seen in Figure 8. Our measurements are performed generally from 9:00 to 16:00 local time. The whole data are averaged per hour during the two years. The diurnal variation shows that

averaged $NH_3$ column increased with temperature in the morning until it peaked at the time interval from



14:00 to 15:00, when temperature reaches the maximum. Then NH$_3$ column reduced with the decrease of air temperature. High temperature promotes the volatilization of N-based fertilizer applied to the local cropland, and at the same time, increased temperature favors the phase transition of particulate ammonium to gas ammonia. Further, the scatter plot of NH$_3$ columns with air temperature in spring and autumn season displays relatively high correlation, with correlation coefficients of 0.53 and 0.48 (Fig. 9), respectively. However, there is no clear correlation between NH$_3$ columns and air temperature in the summer and winter season over Hefei from the scatter plots (not shown). In Hefei, the cropland in spring is characterized by the growing season of wheat and early rice, and synthetic fertilizer is applied during this period. The fertilizer application also tends to occur in autumn, when winter wheat is sowed and a second rice crop is growing. The agricultural practice may explain the correlation between NH$_3$ columns and temperature over Hefei over these two seasons, which suggests that agriculture was the main source of ammonia in spring and autumn.

The polar plots of NH$_3$ columns with wind in Figure 10 show that wind direction mainly ranged from 0° to 270° during the measurement period, indicating that wind was mainly from the north, east, south and south-west. High NH$_3$ columns are associated with wind directions from 45° to 180°, corresponding to wind originating from the east and south-east. The Hefei urban area is located to the east and south of the Hefei site, while the observation site is surrounded by wetlands or cultivated lands to the north and west directions. So the high levels of NH$_3$ partly resulted from transport of air from the urban area. NH$_3$ columns greater than $3\times10^{16}$ molec cm$^{-2}$ correspond to wind speeds less than 1.65 ms$^{-1}$, while wind speeds beyond 1.65 ms$^{-1}$ correspond to NH$_3$ columns below $3\times10^{16}$ molec cm$^{-2}$. This result reflects the well-known phenomenon that large wind speeds increase the mixing and dispersion of the air mass, diluting the concentration of pollution gases. A correlation between NH$_3$ column and relative humidity in air is not observed. Overall, the results indicate that air temperature, wind direction and wind speed are the main factors that influence gaseous NH$_3$ concentrations in Hefei.

In order to get an insight into the potential source regions influencing NH$_3$ concentrations over the Hefei site during the observation period, we ran the HYSPLIT model to calculate the back trajectories of air masses and the potential source contribution function (PSCF) of NH$_3$ columns in different seasons. 24-hour backward trajectories were calculated at 01:00 UTC (8:00 am local time) and 02:00 UTC (9:00 am local time) per day, starting at an altitude of 500 m (approximately 950 hPa). Individual back trajectories were then grouped into three clusters. The back trajectories of air masses and the PSCF of NH$_3$ columns

in different seasons are detailed in a supplement of this paper. The calculation results indicate that agriculture is the dominant source of ammonia in spring, autumn and winter, while urban anthropogenic emissions contribute to the high level of $NH_3$ significantly in summer over the Hefei site. The potential source areas influencing the emission of $NH_3$ were distributed in the local area of Hefei, the northern part

of Anhui province, as well as Shangdong, Jiangsu and Henan provinces.

## 4 Conclusions

Atmospheric ammonia plays an important role in formation of fine particulate matter, affecting air quality and climate. Ground-based FTIR observations have great potential to improve our understanding of the spatial distribution and seasonal variations in atmospheric $NH_3$ on regional scales. In this study, the

spatial distribution and temporal variation, seasonal trends, emission sources and potential sources of $NH_3$ are presented based on ground-based remote sensing of $NH_3$ from December 2016 to November 2018.

The characteristics of the retrieved vertical profiles of $NH_3$ show that the concentration peaked near the surface over all four seasons, with the retrievals being most sensitive in the troposphere. The time series

of the $NH_3$ column obtained during two years shows that the FTIR observation captured the seasonal cycle of $NH_3$, and the seasonal variation is in accordance with that in other areas in China, with similar main emission source, agriculture. Further, the $NH_3$ columns show a 22.14 % $yr^{-1}$ annual increase rate during the measurement period over the Hefei site.

To validate the satellite observations of $NH_3$, we made use of our measurements to compare with co-

located IASI satellite data. The comparison results showed that the IASI satellite data are broadly consistent with the ground-based FTIR measurements. The 230 and 264 pairs of matched data for IASI-A and IASI-B, give mean relative differences of 4.51 % and 0.33 %, with standard deviations of 44.44 % and 41.00 %, respectively. The correlation coefficients (R) are 0.86 and 0.78, respectively.

To examine the contribution of traffic to $NH_3$ columns, we analyze the relationship of $NH_3$ columns with

CO surface concentrations. $NH_3$ columns show high correlation with CO concentrations in summer, with the correlation coefficient (R) of 0.77. Because atmospheric CO is regarded as a primary pollutant mainly emitted from vehicles in urban areas, and there is no significant biomass burning source around the Hefei site, thus the close link between $NH_3$ columns and CO concentrations indicates that the elevated $NH_3$





columns in summer are likely to be partly caused by urban emissions from vehicles.

In order to assess the impact of meteorological parameters on the variation of $NH_3$ columns, we analyzed the relationship between them. High correlation of $NH_3$ columns with air temperature is obvious from their diurnal variation during the observation period. Furthermore, there was a clear correlation between $NH_3$ columns and air temperature in spring and autumn over Hefei, with correlation coefficients of 0.53 and 0.48, respectively. The agricultural practice may explain the correlation between $NH_3$ columns and

temperature over Hefei during these two seasons, which suggests that agriculture was indeed the main source of ammonia in spring and autumn. In addition, wind direction and wind speed clearly influenced the gaseous $NH_3$ concentrations over Hefei.

Further, the back trajectories of air masses calculated by the HYSPLIT model confirmed that agriculture was the dominant source of ammonia in spring, autumn and winter, while urban anthropogenic emissions

contributed to the high level of $NH_3$ in summer over the Hefei site. The potential source areas influencing the $NH_3$ columns were distributed in the local area of Hefei, the northern part of Anhui province, as well as Shangdong, Jiangsu and Henan provinces.

Although $NH_3$ is currently not included in China's strict emission control inventory, we need to investigate the spatial distribution and temporal variation of $NH_3$ together with the driving mechanism

behind them to improve air quality. This study helps to identify the emission sources and potential sources that contribute to $NH_3$ columns over Hefei, a highly populated and polluted area. Our findings have potential implications for reduction of $PM_{2.5}$ pollution in the urban atmosphere over Hefei. This is the first time that ground-based FTIR remote sensing of $NH_3$ columns and comparison with satellite data are reported in China. Future work include the comparison of ground-based FTIR data with in-situ

measurement and model simulations, and to estimate regional emissions of $NH_3$ based on the combination of many measurement techniques.

**Data availability**. The data used in this study are available from the author upon request (wwang@aiofm.ac.cn).


**Supplement**. The supplement related to this article is available online at: xxxxx.

**Author contributions**. WW and NJ worked on the $NH_3$ retrieval methods. LC, MVD, and PFC provided the IASI-$NH_3$ data and contributed to the discussion of the paper. CL, YX, and QH helped explain the

results. CS, HZ, YS and HY took part in the FTIR measurements.



**Competing interests**. The authors declare that they have no conflict of interest.

**Acknowledgements**.

We gratefully acknowledge the support of the National Key Technology R&D Program of China
(2018YFC0213201, 2019YFC0214702, 2016YFC0200404, 2017YFC0210002, 2018YFC0213104,
2019YFC0214802 and 2016YFC0203302), the National Natural Science Foundation of China
(41775025, 41722501, 91544212, 51778596, 41575021 and 41977184), the Major Projects of High
Resolution Earth Observation Systems of National Science and Technology (05-Y30B01-9001-19/20-3),
the Strategic Priority Research Program of the Chinese Academy of Sciences (XDA23020301), the
National Key Project for Causes and Control of Heavy Air Pollution (DQGG0102 and DQGG0205), and
Natural Science Foundation of Guangdong Province (2016A030310115). The authors also gratefully
acknowledge the NOAA Air Resources Laboratory (ARL) for the provision of the HYSPLIT transport
and dispersion model and READY website (http://www.ready.noaa.gov) used in this publication. L.C.
and M.V.D are respectively research associate and postdoctoral researcher with the Belgian F.R.S-FNRS.

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





Table 1. Random and systematic uncertainties used in the error estimation.

| Parameter | Random uncertainty | Systematic uncertainty |
|---|---|---|
| Temperature | 2 K troposphere | 2 K troposphere |
| | 5 K stratosphere | 5 K stratosphere |
| Solar line shift | 0.005 cm$^{-1}$ | 0.005 cm$^{-1}$ |
| Solar line strength | 0.1 % | 0.1 % |
| Solar zenith angle | 0.025° | 0.025° |
| Phase | 0.001 rad | 0.001 rad |
| Zero level shift | 0.01 | 0.01 |
| Wavenumber shift | 0.001 cm$^{-1}$ | 0.001 cm$^{-1}$ |
| Background slope | 0.001 cm$^{-1}$ | 0.001 cm$^{-1}$ |
| Background curvature | 0.001 cm$^{-1}$ | 0.001 cm$^{-1}$ |
| Field of view | 0.001 | 0.001 |
| Line intensity | | 10.0 % |
| Line T broading | | 10.0 % |
| Line P broading | | 10.0 % |
| Interfering species | 2 % ($H_2O$ profile) | 2 % ($H_2O$ profile) |

Table 2. Typical random and systematic errors for each parameter in the retrieval of $NH_3$.

| Parameter | Random error (%) | Systematic error (%) |
|---|---|---|
| Temperature | 1.78 | 2.61 |
| Solar zenith angle | 0.92 | 0.92 |
| Phase | 0.01 | 0.01 |
| Zero level | | |
| Measurement noise | 0.96 | |
| Interfering species | 0.31 | |
| | | |
| Retrieval parameters | | |
| Background curvature | | |
| Smoothing error | 0.14 | |
| Spectroscopy | | 10.70 |
| | | |
| Subtotal error | 2.56 | 11.13 |
| Total error | 11.42 | |

Table 3 Applied filters to the IASI-$NH_3$ data.

| Parameter | Filter criteria |
|---|---|
| IASI-$NH_3$ retrieval error | None |
| Sign of $NH_3$ column | Positive |
| Cloud cover fraction | ≤10% |
| Profile type | Land |
| Spatial sampling difference | 50km |





| Temporal sampling difference | ≤90min |
|---|---|

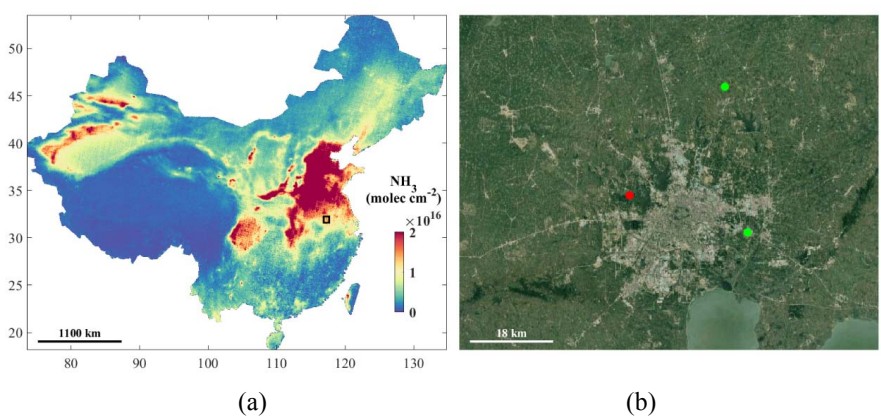

(a)          (b)

Figure 1. Hefei site location. (a) The regional distributions of $NH_3$ columns (molec cm$^{-2}$) from 2008-2018 IASI-A and 2013-2018 IASI-B morning overpasses of ANNI-$NH_3$-v3R data. The rectangle represents the Hefei area. (b) A zoom of the Hefei area (source: © GoogleEarth and Landsat/Copernicus). The red point represents the Hefei site, the green points indicate the location of two point sources identified in Clarisse et al. 2019.

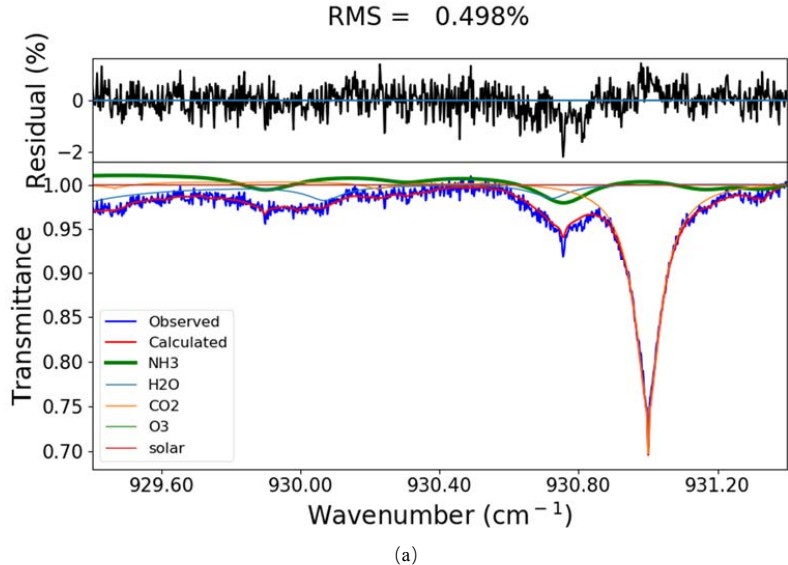

(a)



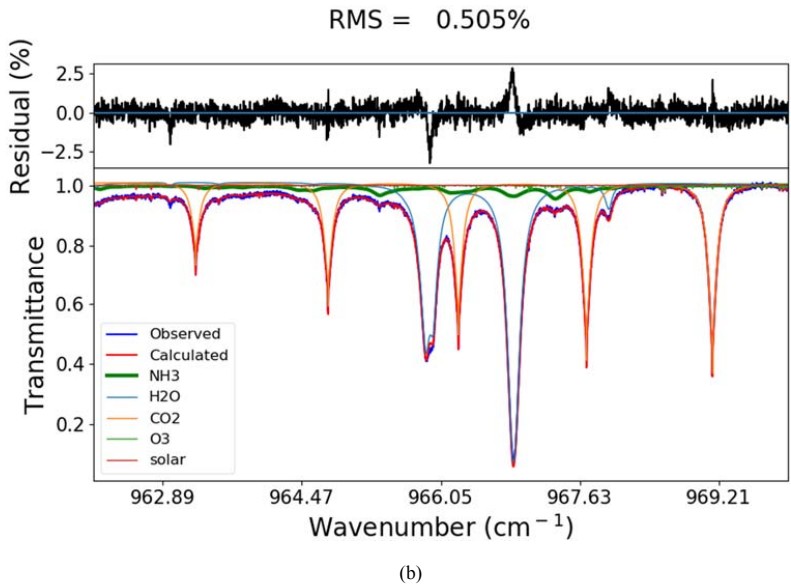

(b)

Figure 2. The spectral windows MW1 (a) and MW2 (b) used for the retrievals of NH₃ at Hefei. An example for a typical measurement is shown (30 August 2018, 10:33 Local time; solar zenith angle: 32.34°; NH₃ total column: $1.64 \times 10^{16}$ molec cm⁻²).

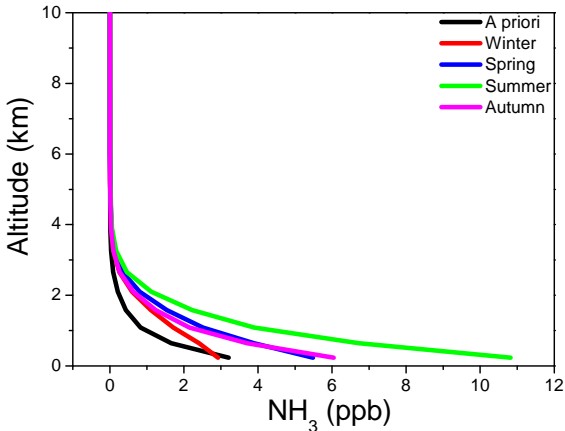

Figure 3. The seasonal averaged vertical profiles of NH₃ (ppb) obtained from the FTIR measurements over Hefei.





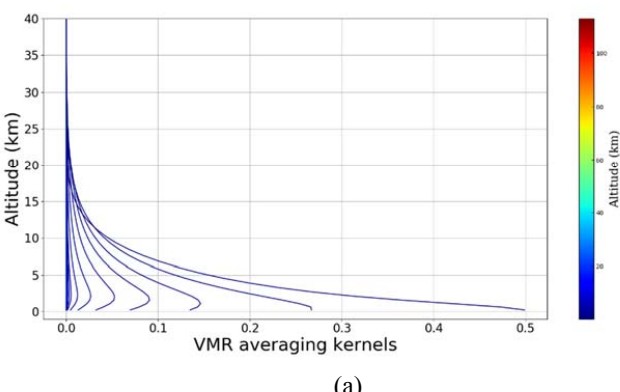

(a)

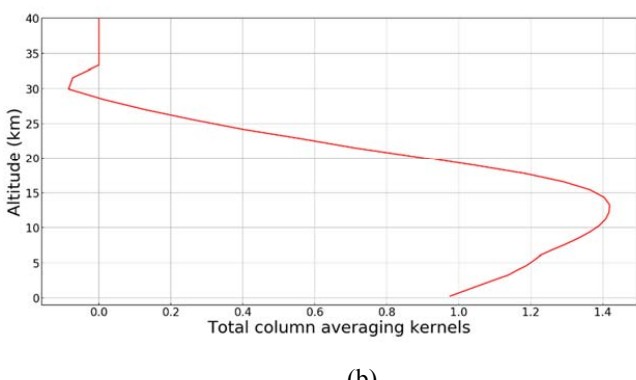

(b)

Figure 4. Typical layer averaging kernels (a) and total column averaging kernel (b) of NH₃.

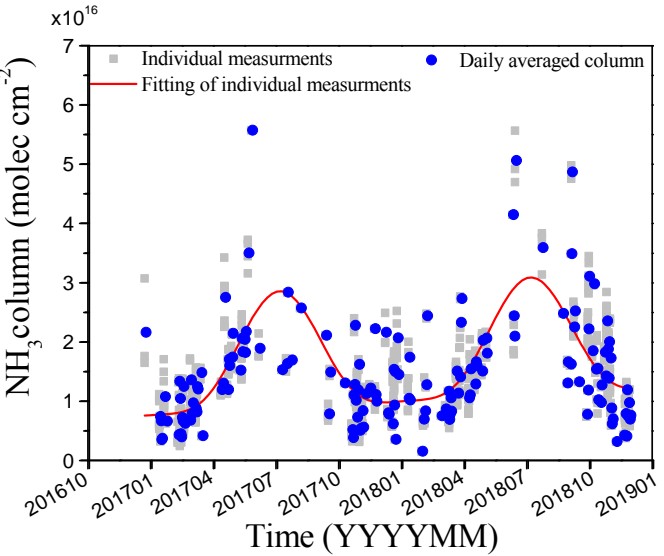





Figure 5. The time series of the ammonia column over Hefei. The grey dots are the individual measurements of NH$_3$; the blue dots represent the daily averaged NH$_3$; the red line is the fitting curve.

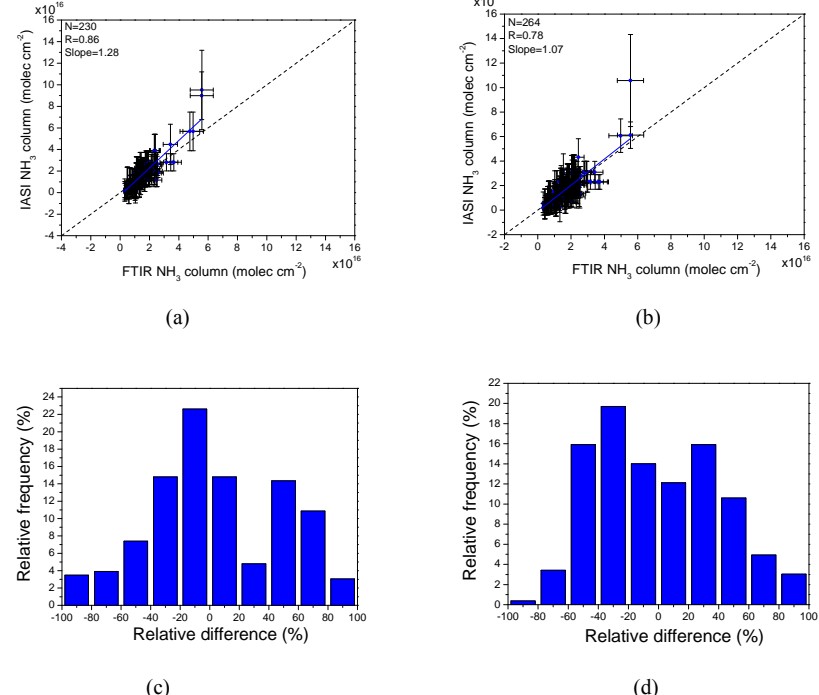

(a)

(b)

(c)

(d)

Figure 6. Retrieved NH$_3$ columns (molec cm$^{-2}$) from ground-based measurement versus IASI-A (a) and IASI-B (b) satellite data. The distribution of the relative difference of NH$_3$ columns for comparison of FTIR with IASI-A (c) and IASI-B (d) satellite data.

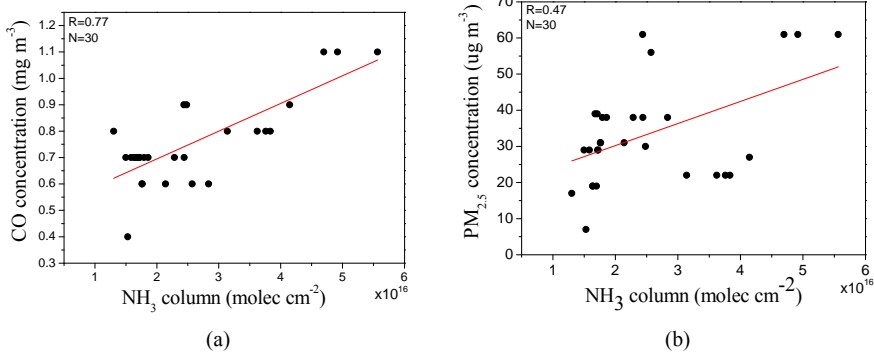

(a)

(b)

Figure 7. Scatter plot of NH$_3$ columns (molec cm$^{-2}$) with CO (mg m$^{-3}$, a) and PM$_{2.5}$ (µg m$^{-3}$, b) concentrations measured at the Dongpu Reservoir air quality monitoring site in summer.





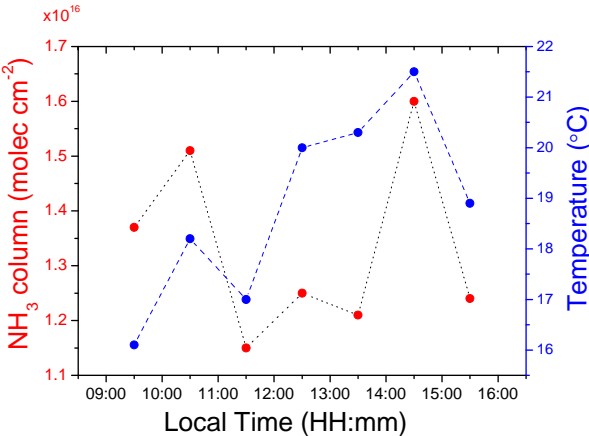

Figure 8. Diurnal variation of NH$_3$ column (molec cm$^{-2}$) in relation to air temperature (°C) during the observation period.

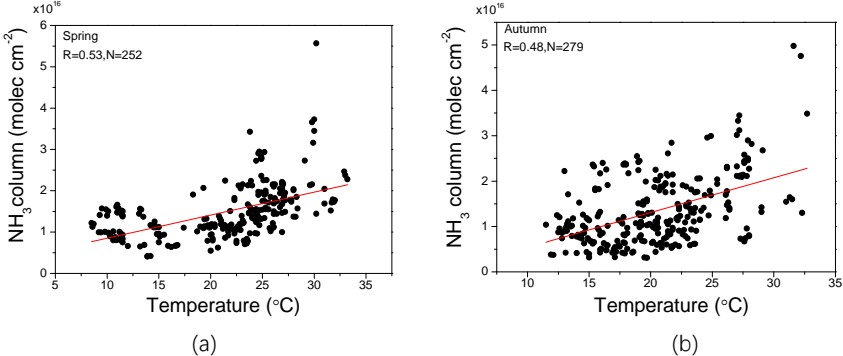

$$\qquad (a) \qquad\qquad\qquad\qquad (b)$$

Figure 9. Scatter plot of NH$_3$ column (molec cm$^{-2}$) with air temperature (°C) in spring (a) and autumn (b).

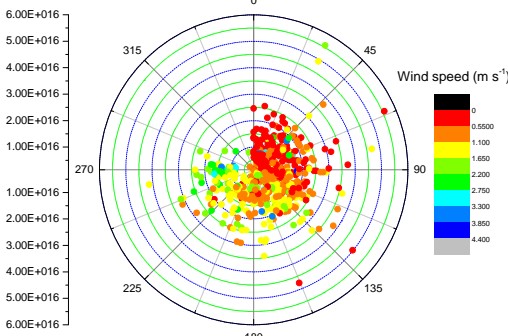

Figure 10. Polar plots of NH$_3$ columns with wind. Radial axes represent the individual NH$_3$ columns (molecule cm$^{-2}$) in relation to wind directions (theta, degrees). The colors denote wind speed (m s$^{-1}$).