# Peer review of "Spatial distribution and seasonal variability in atmospheric ammonia measured from ground-based FTIR observations at"

_Atmospheric Measurement Techniques, 2020_

## Referee Comment (RC1) · Anonymous Referee #1 · 9 Apr 2020

**Overview:**

Wang et al have submitted a manuscript pertaining to ground based infrared-FTIR measurements of ammonia (NH3) at Hefei, China. Two years (Dec 2016 to Nov 2018) of spectral measurements and retrieval processes are detailed, followed with a comparison of the retrieved NH3 total columns to that of co-located NH3 column measurement from the satellite-based IASI instruments. Thereafter the ground-based NH3 columns are used in conjunction with other datasets (in situ carbon monoxide, meteorological data, traffic density) and a back-trajectory modelling study to help attribute NH3 emissions to specific source types. The authors conclude that agricultural NH3 emissions are the main source in spring, autumn and winter, and urban anthropogenic emission (vehicles) dominate in summer. Over the two-years of measurements, an increase of 22.14% in NH3 is reported.

The novelty of this study is that this is the first-time ground based FTIR NH3 measurements at Hefei are reported and compared to satellite data. The high-quality ground based NH3 dataset will be a welcome and important addition to international scientific databases. The manuscript is overall logically structured and well referenced. The writing style is fluent and easily understood. Information on data availability is given and there are no registered conflicts of interest.

Improvements can be made to the manuscript. In its current form I do not recommend publication without major revision to address issues concerning the methodology, trend analysis and the source emission attribution (especially sections 3.4 and 3.5).

The manuscript can be seen as comprised of two parts, that describing the ground based NH3 measurements and satellite comparison, and that of NH3 source attribution using the NH3 column measurements, meteorological & air quality datasets, and back trajectory modelling. The first part is within scope of the AMT journal whereas the NH3 source emission analysis and hypothesis testing in the second part is investigative.

The source emission work is not a simple investigation to support novel measurement validity. To this end, is outside (or bordering on) the scope of AMT. Due to the high level of source attribution investigation and critical issues raised surrounding the analysis of source emission attribution I recommend that this manuscript be either:

1) split into two manuscripts, along the lines of the two parts mentioned previously. The first part submitted to a technical atmospheric journal and the second part submitted to an atmospheric sciences/chemistry journal, or

2) submitted to an atmospheric sciences journal with the focus on the source attribution work.

It is my opinion that the work presented in part 1 requires only minor/moderate changes and technical clarification whereas part 2 requires substantial revision to support the interpretation of the findings or a possible reinterpretation of findings.

Commentary and questions are listed below, and referenced to the documents: amt-2020-39.pdf and amt-2020-39-supplement.pdf

**Specific comments:**

**S1/ Trend analysis.**

It states in the manuscript that there is a 22.14%/yr annual trend in NH3 (L32, L261, L282, L395, L402-L403). This is calculated from the relative difference between two annual means. I think it is a bit premature to calculate an annual trend based on only 2 data points. The difference could be a step function instead of a trend. I recommend that an annual trend is not reported, but the authors may

wish to state the annual means of the two years. It is too speculative at this stage to report a trend, wait for a longer time series.

**S2/ Comparison of ground-based and IASI measurements.**

L294: "Since the IASI-NH3 retrieval does not provide averaging kernels and vertical profiles (Van Damme, et al 2014a), this method for comparison is not applied. Here we therefore compare the IASI satellite and FTIR data directly, without considering the effect of different a priori profiles and averaging kernels."

Such an issue was also confronted by Dammers, et al. 2016., see section 2.3.2. The authors could use the same approach on a subset of the Hefei data, and comment if this alters comparison results. This does assume proxy profiles that are scaled, again see Dammers et al. 2016 for details.

*Dammers, E., et al., An evaluation of IASI-NH3 with ground-based Fourier transform infrared spectroscopy measurements, Atmos. Chem. Phys., 16, 10351–10368, 525 https://doi.org/10.5194/acp-16-10351-2016, 2016.*

**S3/ The descriptive language of correlation coefficients.**

There are inconsistencies in the use of descriptive language used in commentary of correlation coefficients. In the two sentences below, correlations of 0.47 and 0.48 are described as both 'weak' and 'high'.

L345: "NH3 columns show weak correlation (R=0.47) with PM2.5 concentrations"

L359: "Further, the scatter plot of NH3 columns with air temperature in spring and autumn season displays relatively high correlation, with correlation coefficients of 0.53 and 0.48".

Furthermore, at L417: "there was a clear correlation between NH3 columns and air temperature in spring and autumn over Hefei, with correlation coefficients of 0.53 and 0.48, respectively."

I recommend that a standard way of describing the correlations be adopted to achieve consistency. For instance, below is a table from Schober, et al.

**Table. Example of a Conventional Approach to Interpreting a Correlation Coefficient**

| Absolute Magnitude of the Observed Correlation Coefficient | Interpretation |
| --- | --- |
| 0.00–0.10 | Negligible correlation |
| 0.10–0.39 | Weak correlation |
| 0.40–0.69 | Moderate correlation |
| 0.70–0.89 | Strong correlation |
| 0.90–1.00 | Very strong correlation |

*Schober, Patrick, Christa Boer, and Lothar A. Schwarte. "Correlation coefficients: appropriate use and interpretation." Anesthesia & Analgesia 126.5 (2018): 1763-1768.*

Using this example, all the correlations mentioned above would be described as moderate.

**S4/ Calculation of correlation coefficients.**

Could the authors also state the method used to calculate the correlation coefficients, most importantly did the method use uncertainty estimates in both the abscissa and ordinate data points, as both can have large uncertainties.

**S5/ NH3 Air temperature correlation.**

L352: "High correlation of NH3 columns with air temperature is obvious from their diurnal variation during the observation period, as seen in Figure 8."

From figure 8, I could not see a 'high' correlation. The calculated correlation coefficient is also not given. I read data values from figure 8 (and assuming no uncertainty in the read values), then made a scatter plot and calculated the correlation coefficient. See the figure below. I calculated R^2 ~=0.08. This dataset is imprecise and just an indicator. Could the authors please provide the correlation coefficient from the proper dataset and remark if they think the correlation is high. If not, how does this affect the conclusions the authors have made. In my opinion, fig 8 shows a negligible correlation between NH3 and temperature over the analysed averaged time period. There are two spikes in NH3, in the mid-morning and mid-afternoon. Does this correlate with traffic volumes or traffic patterns?

[Figure]

Lastly figure 9 should also include the winter and summer correlation plots, like that for spring and autumn.

**S6/ Section 3.4 and 3.5: NH3 and CO industrial emissions.**

Industrial ammonia:

Throughout the manuscript there is a lack of detail and discussion on possible industrial (point source) emissions of NH3 (such as in Wang, et al, 2015). In section 3.4 analysis only concerns NH3 from vehicle sources. Section 3.5 talks about agricultural emissions (transported into the Hefei region) with a passing mention of "urban anthropogenic emissions" (L388). The only specific mention of point sources is in figure 1. Two large NH3 point sources are identified based on Table A1 of Clarisse et al. 2019.

*Clarisse, L., Van Damme, M.,Clerbaux, C., and Coheur, P.-F.: Tracking down global NH3 point sources with wind-adjusted superresolution, Atmos. Meas. Tech., 12, 5457–5473,* [https://doi.org/10.5194/amt-](https://doi.org/10.5194/amt-) *12-5457-2019, 2019.*

*Wang, S., Nan, J., Shi, C., Fu, Q., Gao, S., Wang, D., Cui, H., Saiz-Lopez, A., and Zhou, B.: Atmospheric ammonia and its impacts on regional air quality over the megacity of Shanghai, China, Sci. Rep., 5,15842, https://doi.org/10.1038/srep15842, 2015.*

Industrial CO emissions:

L329 "To examine the contribution of traffic to NH3 columns, we analyze the relationship of NH3 columns with CO surface concentrations" and L342 "Atmospheric CO is regarded as a primary

pollutant mainly emitted from vehicles in urban areas, and there is no significant biomass burning source around the Hefei site, thus the elevated NH3 columns are likely partly caused by urban emissions from vehicles.". Again, this is a lack of detail on CO industrial (point source) emissions and the effect this would have on any NH3-CO correlations. For example, there is a 700MW coal-fired power station ([https://www.gem.wiki/Hefei-2_power_station](https://www.gem.wiki/Hefei-2_power_station)) to the east of Hefei (~8km), the red flag in the image below, and two power stations in the Hefei region identified in Tan, et al. 2019

[Figure]

Without detailed analysis that includes probable/possible industrial sources of NH3 and CO, conclusions drawn in sections 3.4 and 3.5 are highly speculative and qualitative (using vague phrases such as "likely partly" (L344)).

*Tan, W., Zhao, S., Liu, C., Chan, K., L., Xie, Z., Zhu,Y., Su, W., Zhang, C., Liu, H., Xing, C., and Liu, J., Estimation of winter time NOx emissions in Hefei, a typical inland city of China, using mobile MAX-DOAS observations, Atmos. Environ., 200, 228-242, https://doi.org/10.1016/j.atmosenv.2018.12.009, 2019.*

**S7/ Traffic pollutant correlations.**

L336: "The results show that there exists no correlation between NH3 columns and concentrations of NO2, SO2, PM10 and the value of AQI over the four seasons (not shown)."

Mathematically, there is always a correlation coefficient, if the correlation is less than 0.1, maybe better to state correlation is negligible. Even better, would be to list the correlation coefficient value after each species, i.e. NO2 (XX), SO2 (XX) etc…

It is interesting to read that there is no (negligible) correlation with NO2. NOX (NO2 + NO) is a primary pollutant from vehicle emissions (Zhou et al., 2014, Phillips et al, 2019). What does a NH3-CO moderate correlation with a negligible NH3-NO2 correlation indicate? Could the authors please comment on the reported NH3-NO2 correlation.

*Zhou, Rui, et al. "Study on the traffic air pollution inside and outside a road tunnel in Shanghai, China." PloS one 9.11 (2014).*

*Phillips, F. A., et al.: Vehicle ammonia emissions measured in an urban environment in Sydney, Australia, using open path Fourier Transform Infra-Red Spectroscopy. Atmosphere, 10, 208, [https://doi.org/10.3390/atmos10040208](https://doi.org/10.3390/atmos10040208), 645 2019.*

There is also no quantitative measurand of traffic volumes. If NH3 emissions were primarily (or even partly) from vehicle emissions, then more in-depth quantitative analysis is required.

**S8/ NH3 and PM2.5.**

L345: "Meanwhile, NH3 columns show weak correlation (R=0.47) with PM2.5 concentrations (Fig. 7(b)), meaning that NH3 contributed to the formation of fine particulates significantly in summer.".

This qualitative statement needs expanding. More in-depth analysis of the Hefei dataset is needed. A more quantitative investigation is required into PM2.5 and NH3 correlation and/or causation. It is well known NH3 contributes to PM2.5.

**S9/ Column to point correlations**

The NH3 measurements are partial columns extending from ~0-4km in altitude. The meteorological and air quality datasets are from point source ground level measurements, located a distance from the NH3 measurements. These two datasets will have quite different footprints. The in-situ ground-based measurements will be have a more localised footprint, whereas the column measurements will have a larger regional footprint. Could the authors comment on this fact and how it affects correlation interpretation.

**S10/ Agricultural NH3 emissions**

L362 – 367: "In Hefei, the cropland in spring is characterized by the growing season of wheat and early rice, and synthetic fertilizer is applied during this period. The fertilizer application also tends to occur in autumn, when winter wheat is sowed, and a second rice crop is growing. The agricultural practice may explain the correlation between NH3 columns and temperature over Hefei over these two seasons, which suggests that agriculture was the main source of ammonia in spring and autumn."

This a good starting hypothesis to investigate, but it is left as a conjecture. Too vague language is used, e.g. "May explain" and "suggests". These are good starting points, but quantitative substantial evidence is required. This is lacking. Seasonal (hence temperature) correlations is not a strong causal link without additional collaborating evidence. The authors need to add such evidence to prove the statement in the manuscript.

**S11/ Windrose**

In such wind rose polar plots the wind strength is usually plotted as the radial component and the NH3 abundance as coloured points. I would strongly recommend replotting with wind speed as the radial axis of the polar plot and NH3 concentrations as coloured dots.

Looking at the current plots, the greatest wind speed is less than 4.5ms$^{-1}$. This is classified as a gentle breeze using the Beaufort scale. This indicates that the majority of measured NH3 is from local sources (assuming hourly- low daily NH3 lifetimes). The highest concentrations of NH3 is measured during easterly winds, the direction of the main Hefei urban area and the lowest concentrations from the west, a less populated and dense area. At L368 the authors comment on the wind rose data, but do not use it to help attribute the urban emission sources (traffic vs industrial). High NH3 concentration points are in the NW quadrant, the direction of an NH3 point source shown in figure 1. There is no commentary on this fact.

**S12/ NH3 lifetime.**

There is only passing mention of NH3 lifetime (L55). Is there an expected difference in NH3 lifetime depending on emission source (i.e. rural vs industrial) and would this influence NH3 source attribution?

**S13/ Back-trajectory calculations using HYSPLIT.**

At L380 the authors introduce a new method to help assess NH3 source emissions. Back trajectory modelling is introduced, but I cannot understand why the bulk of the findings are relegated to the supplement. I recommend that the supplement content and the main manuscript from L380 -390 be concatenated into a new section, e.g. section 3.6.

In addition, a much more detailed explanation of the model set up (meteorological fields, resolution, cluster analysis details, etc) and analysis (source attribution) needs to be undertaken. Example: the PSCF's in figure S2 are hard to see in detail and require further inquiry. The interpretation of the results at L386-390, and the supplement, need more rigorous assessment in a scale more regional to Hefei (due to low wind speeds). There is no mention of carbon monoxide transport either, and the effect this would have on NH3-CO correlations and source attribution.

**S14/ Concluding statements in section 4.**

This section requires major revision based upon the issues raised in this review.

Currently there is not enough collaborating evidence in the manuscript to explicitly state with certainty the origin and abundance of NH3 emissions measured at Hefei.

**Minor/Technical:**

M1/ L30: "emission sources and potential sources". What is the difference between emission and potential sources? Do you mean known emission sources and unknown emission sources? Can you please make this clearer and correct through-out the manuscript?

M2/ L34: "The results demonstrate the IASI data are in broad agreement with our FTIR data.". I think it best to rephrase as "The results demonstrate that the IASI and FTIR data, over Hefei, are in broad agreement.".

M3/ L64: "This is mainly caused by lack of representative measurements of atmospheric NH3.". Could a reference to this statement be added?

M4/ $NH_4^+$: There is no mention of $NH_4^+$ with respect to NH3 partition equilibrium in NHx, and how this will affect the measured NH3 abundances. Could the authors please comment on this aspect. Lachatre, et al, 2019 would be a good starting point.

M5/ L68: There are ~12 references to a single statement. Can this be cut down to only the essential references needed. Passive sampling is not the focus of this manuscript.

M6/ L71: Pedantic but…could 'Sticky' be replaced with a more scientific descriptive term, or place in quotation i.e. 'sticky'.

M7/ L78: Replace 'ground-based' with 'ground level'

M8/ L120: "Furthermore, ground based observations of NH3 have been sparse throughout China (Liu et al., 2011; Xu et al., 2015)." Ground based or ground level in situ?

M9/ L132: Replace "reduces" with "reductions of"

M10/ L145: I think "Materials" is the incorrect word, maybe "Measurements" would be better.

M11/ L164: "liquid-nitrogen-cooled MCT/InSb detector". Just to clarify, is this a single detector or dual HgCT and InSb detectors?

M12/ L165: Replace 'suit' with 'suite'

M13/ L171: "At the same time, the indoor pressure, temperature and relative humidity are logged continuously.". Are these important parameters for the FTIR measurements?

M14/ L173: "similar to the NH3 retrieval strategies in Dammers et al. (2015)". What are the differences? Are the Microwindows the same?

M15/ L181: "A priori profiles of NH3 and interfering gases are taken from the Whole Atmosphere Community Climate Model (WACCM, v.6_120_99) in combination with initial measurement values."

There is no NH3 modelling output from WACCMv6. Could the authors please correctly reference the NH3 apriori source used.

Is a static apriori used? Or does it change throughout the retrieval time series?

M16/ L181: "…in combination with initial measurement values.". Sorry, I do not understand this. Can you please explain what initial measurements are, and how they are combined with (so-called) WACCMv6 profiles to obtain an apriori?

M17/ L182: "WACCM, v.6_120_99". I think this can be referenced better. '_120_99' is not part of the WACCM simulation identify code. Maybe better to reference it this way: (WACCMv6, Garcia et al. 2007).

*Garcia, R. R.,Marsh, D. R., Kinnison, D. E., Boville, B. A., and Sassi, F.: Simulation of secular trends in the middle atmosphere, 1950-2003, 20 J. Geophys. Res., 112, D09301, doi:10.1029/2006JD007485, 2007.*

M18/ L183: "for ammonia was constructed to be diagonal, with standard deviations of 100% for all layers." What was the basis for assuming an uncertainty of 100% and is there any inter-layer covariance/correlation constraint?

M19/ I would also encourage the authors to include another sentence (or two) after the description of the apriori profile and uncertainties describing other state vector retrieval parameters such as phase fitting, instrument line shape, continuum fit, zero level offset etc.

M20/ Also…could be authors also state the SNR (or $S_e$) of the measurements, and either if the SNR is held static or varies on a spectrum by spectrum basis.

M21/ How is the H2O apriori set? For instance, is it from NCEP daily profiles, or from WACCM, or from a previous H2O retrieval?

M22/ L190: "about 0.498 % and 0.505%", could these values be rounded to the correct number of significant figures.

M23/ Fig 2 and L188: "The measured spectrum is shown in blue, the fitted spectrum in red and the residual in black". There are two blue lines, 'observed' and H2O. Either state as 'dark blue' or switch colour.

M24/ I recommend that section 3.1.2 lines L238 to 246 be moved to the end of section 2.2. This is because it details retrieval information content and logically comes after spectral fitting.

M25/ L239: "It is evident that the retrieval is most sensitive to the troposphere, where the concentration of ammonia peaks." Add in 'lower': "It is evident that the retrieval is most sensitive in the lower troposphere, where the concentration of ammonia peaks"

M26/ L243: "The degrees of freedom for signal (DOFS) value is 1.10 given by this measurement". Can the authors indicate at what height, from the surface up, does the DOFS reach 1.0, essentially showing the partial column range in which there is a signal?

M27/ Fig 4. The individual layer averaging kernels are all the same colour, very confusing. Maybe only plot for layers 0-10km. There are also no units for the VMR averaging kernels (I assume VMR/VMR)? Also, for the total column averaging kernel, what are the units?

M28/ Section 3.1.1 could be transferred to be section 2.3. (IASI becomes section 2.4). It makes more sense that the entire retrieval process from retrieval set up to retrieval output then error analysis is all together in one section.

M29/ L215: "The error calculation is based on attributing uncertainties to all parameters used in the profile retrieval". Maybe better stated as "The error calculation is based on attributing uncertainties to all parameters in the retrieval state vector".

M30/ L221: "Table 1 lists the uncertainties of the parameters assumed in the retrievals." Maybe better stated as: "Table 1 lists the uncertainties of the retrieval state vector parameters."

M31/ L222: "The total errors are about 11.42 % based on the combination of random and systematic errors". Is error analysis applied to each individual retrieval? If so, the mean total error and 1-sigma standard deviation of the time series could be given. This would be a better indication of the overall timeseries uncertainty than a single retrieval.

M32/ Table 1:

Could the authors please state where or how the temperature profile uncertainties were set/calculated.

Did the authors calculate the uncertainty associated with inferring species spectral uncertainties? i.e. the line intensity (and pressure and temperature broadening) uncertainties in $CO_2$, $H_2O$ etc…

Could 'NH3' be put in front of Line Intensity, and P, T broadening (broadening, not broading). Could the authors also reference the origin of the 10% spectral uncertainties.

M33/ Section 3.1.2 and Fig 3: With Section 3.1.1 moved to section 2, along with the second half of section 3.1.2, this leaves a single paragraph (L230-237) in section 3.1.2. This paragraph could be moved into section 3.2. There is very little vertical information due to the low DOFs, basically profile scaling.

Figure 3 offers little information due to the low DOFs. The variation within a season is not presented so it is hard to conclude if the seasonal differences in profile are significant. Maybe reconstruct figure 3 to have 4 plots (2 by 2), one for each season and for each season include the spread (1-sigma SD?) at each level.

M34/ L235: "The seasonal averaged surface level of NH3 decreased from 10.82 ppb in summer to 2.92 ppb in winter during 2017 and 2018, and the corresponding values are about 5.48 and 6.04 ppb in spring and autumn, respectively.". It would be advantageous if for each season the mean and spread (1-sigma SD) was given.

M35/ L250: "Many spectra ranging from 700 to 1350 cm-1 are saturated in summer (due to high humidity), causing the retrieved NH3 data to be sparsely sampled relative to those in other seasons". MW1 (~929.6-931.2cm-1) (fig 2a) shows small H2O absorption features, as opposed to MW2. In summertime, could plausible retrievals be performed with only using MW1 without altering the DOFS too much? This may help increase the number of observations.

M36/ L276: "It follows therefore that seasonal variation of NH3 columns in the Hefei area accords with that in other areas in China, with the main emission source being agriculture." This sentence is not needed as a near repeat of the statement at L263.

M37/ L287: Maybe rewritten as "Here we present a comparison of the ground-based NH3 column measurements with the IASI satellite measurements".

M38/ L290. How was the co-location criteria of 0.5 degrees and 90 minutes decided upon?

M39/ L303: "The Relative differences larger than 100% were considered as outliers from the data". On what basis, and why the value of 100%? Was the ground based or satellite measurements anomalous?

M40/ L305: "with standard deviation of 44.44% and 41.00%, respectively.". Maybe 44% and 41%.

M41/ L308: "Additionally, the distributions of the relative difference of the two dataset show that, the relative bias mainly range from -60% to 80% for IASI A data, from -60% to 60% for IASI B data, and the bias from -20% to 0% as well as -40% to -20% has the highest frequency in both respective bins (Figure 6 (c) and (d))."

Bias refers to the overall difference between the two datasets, so I think it is better worded as:

"Additionally, the distributions of the relative difference of the two datasets show that the relative differences of a measurement pair, mainly range from -60% to 80% for IASI A data, from -60% to 60% for IASI B data as seen in figure 6 (c) and (d)."

M42/ L313L First time the acronym NDACC is used. For the first instance can the full title please be used. Also maybe better stated as: "nine NDACC FTIR stations".

M43/ L317: "The correlation coefficient (R) between the CrIS NH3 total columns and FTIR data was 0.77, and the relative difference is 0-5 % with a standard deviation of 25–50 % for comparison of high levels of NH3."  Sorry I do not understand the '0-5%' statement, can you please explain it more clearly, how can you have multiple mean differences?

M44/ L320: "results from other NDACC site data". Is Hefei a NDACC affiliated FTIR site? If not, then the current wording can be read that it is. Maybe replace with "comparison results in the aforementioned literature"

M45/ L330: "The Dongpu Reservoir air quality monitoring station (31.91°N, 117.16°E) is very close to our site, part of a National Ambient Air Quality Monitoring Network, which monitors and routinely publishes the concentrations of main gaseous pollutants, including CO, NO2, PM2.5, PM10, SO2, O3 and Air Quality Index (AQI) etc". Could the distance from the air quality station to the FTIR site please be stated and a reference added pertaining to the National Ambient Air Quality Monitoring Network, (esp. for the Hefei site if possible). Could the Dongpu site please be marked in figure 1b.

M46/ L376: "Overall, the results indicate that air temperature, wind direction and wind speed are the main factors that influence gaseous NH3 concentrations in Hefei.". What about source emissions?

M47/ L393: the adjective 'great' can be omitted.

M48/ L415: replace "meteorological parameters" with "meteorological conditions".

M49/ L432: "This is the first time that ground-based FTIR remote sensing of NH3 columns and comparison with satellite data are reported in China". A valid and important finding and should be reported at the start of the conclusion section.

M50/ L405: "To validate the satellite observations…". In this case since the satellite measurements have been compared against other measurements favourably, the satellite measurements are used to confirm/validate the Hefei site ground base measurements.

M51/ Fig 1: Could longitude and latitude units please be put on both axis. The area displayed could also be reduced to focus on the eastern and middle region of China. In map B, could the Dongpu AQ station also be identified.  Could the location identifier 'dots' also be bigger in both maps. Could a distance scale also be put on the map?

Caption: "The regional distributions of NH3 columns (molec cm-2) from 2008-2018 IASI-A and 2013-2018 IASI-B morning overpasses of ANNI-NH3-v3R data". Are these annual mean columns? Good idea to specify the temporal period.

M52/ Fig 2a and b: the title consists of RMS. If possible, add in the DOFS and CHI^2 fit variables as well for completeness.

M53/ Fig 5: Could 1-sigma standard deviation spread/error bars be put on the daily averages.

M54/ Fig 6: A lot of white space in plots A &B, could the X-axis range be shortened, MAX X ~= 8e16.

M55/ Fig 7: Could the uncertainty bars be added for each point, like that in fig 6 a and b.

Caption: "Scatter plot of NH3 columns (molec cm-2) with CO (mg m-3, a) and PM2.5 (µg m-3, b) concentrations measured at the Dongpu Reservoir air quality monitoring site in summer." Are these daily averages? Please add in temporal period.

M56/ Fig 9:  Caption: "Scatter plot of NH3 column (molec cm-2) with air temperature (°C) in spring (a) and autumn (b)." Again, please add temporal period (hourly, daily, monthly…).

M57/ Fig S1 and S2, it is hard to ascertain information from both plots. As a first step, could the map lat/lon boundaries be reduced to concentrate on areas of interest better. I.e. zoomed in a bit.

M58/

Franco, B. Clarisse, L. , Stavrakou, T., Müller, J.-F, Van Damme, M. ,Whitburn, S., Hadji-Lazaro, J. , Hurtmans, D., Taraborrelli, D. , Clerbaux,C. , Coheur,P.-F.: A General Framework for Global Retrievals of Trace Gases From IASI: Application to Methanol, Formic Acid, and PAN, J. Geophys. Res.-Atmos., 123, 963-984, https://doi.org/10.1029/2018JD029633, 2018.

Is this reference relevant?

---

## Referee Comment (RC2) · Anonymous Referee #2 · 27 Apr 2020

General comments

This paper presents ground based FTIR measurements at the Hefei site including error analysis of the NH$_3$ retrievals, vertical distribution, time series and seasonal trend analyses. More additional works such as comparisons of IASI data, relationship with surface CO, temperature, wind speed and direction, and back trajectories analysis are made in this paper. I believe this paper is suitable for publication to Atmos. Meas. Tech. after considering comments as below.

As for dividing into two papers by anther refree, if authors can prepare more analyses and discussions for AMT and another paper, I think it would be better. If not, One paper of AMT looks not bad.

Specific comments

line 2

I feel "measured" is not fitted because of "measured from observations". Retrieved, derived, obtained etc. would be better.

Line 32 and 402

If possible, could you provide error (one standard deviation) for 22.14 % yr-1 annual increase rate?

Line 38-39

"Further, high correlation of NH$_3$ columns with air temperature is obvious from their diurnal variation during the observation period."

"In addition, the clear correlation between NH$_3$ columns and air temperature in spring and autumn over Hefei, suggests that agriculture was indeed the main source of ammonia in spring and autumn."

I think a correlation coefficient NH$_3$ columns with air temperature should be provided, since NH$_3$ columns with CO concentration is described as R=0.77.

Line 83-89

Authors had better add the GOSAT retrieval from TANSO-FTS TIR spectra as recent results.

Citation: Someya, Y., Imasu, R., Shiomi, K., and Saitoh, N.: Atmospheric ammonia retrieval from

the TANSO-FTS/GOSAT thermal infrared sounder, Atmos. Meas. Tech., 13, 309–321, https://doi.org/10.5194/amt-13-309-2020, 2020.

Line 99-102

"More recently, FTIR measurements have been shown to also provide total column and vertical profiles of ammonia at a high temporal resolution, and are now also used for validation of satellite $NH_3$ observations (Dammers, et al., 2015; Dammers, et al., 2016; Dammers, et al., 2017b)."

I recommend to adding a name of the satellite, that is, IASI.

Line 226

Some of working in same field can understand "phase", but for wider readers, a little explanation might be necessary.

Line 250

"Many spectra ranging from 700 to 1350 $cm^{-1}$ are saturated in summer (due to high humidity), causing the retrieved $NH_3$ data to be sparsely sampled relative to those in other seasons."

Are there else any better spectral windows for retrieval of $NH_3$ in summer season?

Line 281

"The annual increasing rate of ammonia columns in Hefei estimated by our two-year FTIR measurements (22.14 % $yr^{-1}$) is much larger than the reported value by satellite observations over China. This is likely due to the different sampling years. The increasing trend of $NH_3$ in Hefei is likely caused by either an increased fertilizer use, or increasing air temperature, or decreased sulfur emissions due to strict $SO_2$ control measures."

Is it possible to verify the annual increasing rate of ammonia columns over Hefei using other satellite or model data?

Line 291

"We remove the data with negative IASI-$NH_3$ columns due to large retrieval error."

Negative values for the IASI-$NH_3$ columns are not physically meaning. I think large retrieval error is not fundamental reason.

Line 303-311

There would be different results comparing of IASI A and IASI B data with the Hefei FTIR data, but they are within one standard deviation. Could you describe this reason by citing the literatures or technical reports from the IASI team? If impossible, a description that the difference is within one standard deviation would be there.

Line 319-321

"So the relative differences between IASI total columns and our FTIR data and standard deviations of the differences are within the range of comparison results from other NDACC site data, and the correlation coefficients are comparable to that of other comparison results."
In table 3 of Dammers et al. (2017b) paper, the mean relative difference (MRD) at Wollongong site is only positive (6.0 ± (74.3)%) and other sites are negative. If readers know this, readers may get confused. More detail discussions and descriptions are necessary.

Line 338

"However, $NH_3$ columns show high correlation with CO concentrations in summer, as displayed in Figure 7(a)."
How is other seasons ?

Line 345

"Meanwhile, $NH_3$ columns show weak correlation (R=0.47) with $PM_{2.5}$ concentrations (Fig. 7(b)), meaning that $NH_3$ contributed to the formation of fine particulates significantly in summer."
If so, a correlation with $PM_{2.5}$ concentrations in other seasons might be higher than that in summer. Did authors check them?

Line 352

"High correlation of $NH_3$ columns with air temperature is obvious from their diurnal variation during the observation period, as seen in Figure 8. Our measurements are performed generally from 9:00 to 16:00 local time. The whole data are averaged per hour during the two years."
Considering discussions that follow, a plot in Figure 8 prepared for whole data (I understand all seasons) should be prepared for each seasons. Could authors explain a reason that $NH_3$ columns decreased from 11:30 to 13:30 in figure 8? If plots are prepared for each season, decreasing in spring and autumn might be appeared.

Technical corrections

line 28-29

I fell there is a duplication. One idea is to remove "a measurement site in".

Line 32

measurement-> measurements

Line 35

Analyze -> analyzed?

Line 143

Remove "retrieved"

Line 169

"recorded"-> "has been recording"

Line 179

"vertical profile"->" vertical profiles"

Line 204

"general"-> "generally"

Line 235

"The seasonal averaged surface level of $NH_3$ decreased from 10.82 ppb in summer to 2.92 ppb in winter during 2017 and 2018, and the corresponding values are about 5.48 and 6.04 ppb in spring and autumn, respectively."

Line 261

"The annual mean NH3 column is $1.31 \times 10^{16}$ and $1.60 \times 10^{16}$ molec $cm^{-2}$, respectively, with an increase rate of about 22.14 %."

If readers can know errors or standard deviation, they might be good.

Line 243, 244, 249 and other

"retrievals at the Hefei site" or "retrievals in the Hefei site."    A lot of inconsistency, "at the Hefei site" might to be good.

Line 264

If authors use "practices" as a noun, I think "maybe" is adverb and there is no verb in this sentence.

Line 269

"agriculture"-> agricultural area?

Line 325

"tunnel studies"

Simple description for them is grateful.

Line 330

"The Dongpu Reservoir air quality monitoring station (31.91°N, 117.16°E) is very close to our site, part of a National Ambient Air Quality Monitoring Network, which monitors and routinely publishes the concentrations of main gaseous pollutants, including CO, $NO_2$, $PM_{2.5}$, $PM_{10}$, $SO_2$, $O_3$ and Air Quality Index (AQI) etc."

There is no citation for the data.

Line 434

"Future work" -> "Future works"

Line 435

"to estimate regional"->"estimating regional" or "estimation of regional"

Figure 1 caption

"The regional distributions of NH3 columns (molec $cm^{-2}$) from 2008-2018 IASI-A and 2013-2018 IASI-B morning overpasses of ANNI-NH3-v3R data."

Are they averaged values or overlaid ?    Clarification would be necessary.

Figure 4 (a)

Higher than 40 km should be removed for a color bar for the altitude and replot would be necessary. If Authors can do them, readers may understand which colors are altitudes for VMR averaging kernels. But I don't know it is useful.

Figure 7 (b)

"ug"->"micro g" Micro is small Greek letter.

Figure 10

Digits after the decimal point might be not necessary for $NH_3$ column. Also digits after the first decimal point might be not necessary for wind speed.

I feel wind speed would for radial axes and $NH_3$ columns for color bars would be better for better understanding relationship of $NH_3$ columns to wind direction and speed. If authors did not try, please try.

Figure S1.

Back trajectories colored with black are very cloudy, if possible could authors color them for each cluster?

Figures S1 and S2

What is light blue curved lines? There is no description for them.

---

## Author Comment (AC1) · 8 Jul 2020

**Response to comments #1**

We appreciate your constructive and positive comments. The comments and proposed corrections have been taken into account and helped improving the paper. Each comment has been addressed as follows. There is an extensive discussion among the authors regarding how to revise the content. So the response is delayed, and we are sorry for this.

**Overview:**

Wang et al have submitted a manuscript pertaining to ground based infrared-FTIR measurements of ammonia (NH3) at Hefei, China. Two years (Dec 2016 to Nov 2018) of spectral measurements and retrieval processes are detailed, followed with a comparison of the retrieved NH3 total columns to that of co-located NH3 column measurement from the satellite-based IASI instruments. Thereafter the ground-based NH3 columns are used in conjunction with other datasets (in situ carbon monoxide, meteorological data, traffic density) and a back-trajectory modelling study to help attribute NH3 emissions to specific source types. The authors conclude that agricultural NH3 emissions are the main source in spring, autumn and winter, and urban anthropogenic emission (vehicles) dominate in summer. Over the two-years of measurements, an increase of 22.14% in NH3 is reported.

The novelty of this study is that this is the first-time ground based FTIR NH3 measurements at Hefei are reported and compared to satellite data. The high-quality ground based NH3 dataset will be a welcome and important addition to international scientific databases. The manuscript is overall logically structured and well referenced. The writing style is fluent and easily understood. Information on data availability is given and there are no registered conflicts of interest.

Improvements can be made to the manuscript. In its current form I do not recommend publication without major revision to address issues concerning the methodology, trend analysis and the source emission attribution (especially sections 3.4 and 3.5).

The manuscript can be seen as comprised of two parts, that describing the ground based NH3 measurements and satellite comparison, and that of NH3 source attribution using the NH3 column measurements, meteorological & air quality datasets, and back trajectory modelling. The first part is within scope of the AMT journal whereas the NH3 source emission analysis and hypothesis testing in the second part is investigative.
The source emission work is not a simple investigation to support novel measurement validity. To this end, is outside (or bordering on) the scope of AMT. Due to the high level of source attribution investigation and critical issues raised surrounding the analysis of source emission attribution I recommend that this manuscript be either:
1) split into two manuscripts, along the lines of the two parts mentioned previously. The first part submitted to a technical atmospheric journal and the second part

submitted to an atmospheric sciences/chemistry journal, or
2) submitted to an atmospheric sciences journal with the focus on the source attribution work.

It is my opinion that the work presented in part 1 requires only minor/moderate changes and technical clarification whereas part 2 requires substantial revision to support the interpretation of the findings or a possible reinterpretation of findings. Commentary and questions are listed below, and referenced to the documents: amt-2020-39.pdf and amt-2020-39-supplement.pdf

**Specific comments:**

S1/ Trend analysis.

It states in the manuscript that there is a 22.14%/yr annual trend in NH3 (L32, L261, L282, L395, L402-L403). This is calculated from the relative difference between two annual means. I think it is a bit premature to calculate an annual trend based on only 2 data points. The difference could be a step function instead of a trend. I recommend that an annual trend is not reported, but the authors may wish to state the annual means of the two years. It is too speculative at this stage to report a trend, wait for a longer time series.

Response: We removed the descriptions about the annual trend throughout the paper, and state the annual means of the two years and the increasing value between the two years.

S2/ Comparison of ground-based and IASI measurements.

L294: "Since the IASI-NH3 retrieval does not provide averaging kernels and vertical profiles (Van Damme, et al 2014a), this method for comparison is not applied. Here we therefore compare the IASI satellite and FTIR data directly, without considering the effect of different a priori profiles and averaging kernels."

Such an issue was also confronted by Dammers, et al. 2016., see section 2.3.2. The authors could use the same approach on a subset of the Hefei data, and comment if this alters comparison results. This does assume proxy profiles that are scaled, again see Dammers et al. 2016 for details.

Dammers, E., et al., An evaluation of IASI-NH3 with ground-based Fourier transform infrared spectroscopy measurements, Atmos. Chem. Phys., 16, 10351–10368, 525 https://doi.org/10.5194/acp-16-10351-2016, 2016.

Response: We applied the FTIR averaging kernels to IASI profile to account for the effects of the a priori information and vertical sensitivity of the FTIR retrieval, following the method in Dammers et al. (2016), then computed the total column using the smoothed IASI profile. We added the comments if this alter comparison results in section 3.2 "Comparison with satellite data."

S3/ The descriptive language of correlation coefficients.

There are inconsistencies in the use of descriptive language used in commentary of correlation coefficients. In the two sentences below, correlations of 0.47 and 0.48 are described as both 'weak' and 'high'.

L345: "NH3 columns show weak correlation (R=0.47) with PM2.5 concentrations"

L359: "Further, the scatter plot of NH3 columns with air temperature in spring and autumn season displays relatively high correlation, with correlation coefficients of 0.53 and 0.48".

Furthermore, at L417: "there was a clear correlation between NH3 columns and air temperature in spring and autumn over Hefei, with correlation coefficients of 0.53 and 0.48, respectively."

I recommend that a standard way of describing the correlations be adopted to achieve consistency. For instance, below is a table from Schober, et al.

| Table. Example of a Conventional Approach to Interpreting a Correlation Coefficient | |
| --- | --- |
| **Absolute Magnitude of the Observed Correlation Coefficient** | **Interpretation** |
| 0.00–0.10 | Negligible correlation |
| 0.10–0.39 | Weak correlation |
| 0.40–0.69 | Moderate correlation |
| 0.70–0.89 | Strong correlation |
| 0.90–1.00 | Very strong correlation |

Schober, Patrick, Christa Boer, and Lothar A. Schwarte. "Correlation coefficients: appropriate use and interpretation." Anesthesia & Analgesia 126.5 (2018): 1763-1768. Using this example, all the correlations mentioned above would be described as moderate.

Response: we modified the descriptive language of correlation coefficients throughout the paper, according to the way of describing the correlations in Schober et al.(2018).

S4/ Calculation of correlation coefficients.

Could the authors also state the method used to calculate the correlation coefficients, most importantly did the method use uncertainty estimates in both the abscissa and ordinate data points, as both can have large uncertainties.

Response: We added the description about the the method used to calculate the correlation coefficients, and used uncertainty estimates in both the abscissa and ordinate data points in the third paragraph in section 3.2 "Comparison with satellite data".

S5/ NH3 Air temperature correlation.

L352: "High correlation of NH3 columns with air temperature is obvious from their diurnal variation during the observation period, as seen in Figure 8."

From figure 8, I could not see a 'high' correlation. The calculated correlation coefficient is also not given. I read data values from figure 8 (and assuming no uncertainty in the read values), then made a scatter plot and calculated the correlation coefficient. See the figure below. I calculated $R^2 \sim =0.08$.

This dataset is imprecise and just an indicator. Could the authors please provide the correlation coefficient from the proper dataset and remark if they think the correlation is high. If not, how does this affect the conclusions the authors have made. In my opinion, fig 8 shows a negligible correlation between NH3 and temperature over the analysed averaged time period. There are two spikes in NH3, in the mid-morning and

mid-afternoon. Does this correlate with traffic volumes or traffic patterns?

Response: We calculated the correlation coefficient between NH$_3$ columns with air temperature from their diurnal variation, and the correlation coefficient is 0.39, showing a weak correlation between NH$_3$ and temperature from their diurnal variation. The NH$_3$ data measured after 14:00 PM local time are sparse as we collect the MIR and NIR solar spectra alternately every day, so the data in some time period lacks of representation for diurnal variation analysis. Therefore we removed the discussion about the correlation of NH$_3$ columns with air temperature from their diurnal variation, including Figure 8, in section 3.3 "Identification of emission sources of NH$_3$".

I think the two spikes in NH$_3$ columns in the mid-morning and mid-afternoon, is not correlated with traffic patterns, as they should occur on 7:00-9:00 AM and and 17:00-19:00 PM if they are correlated with traffic patterns.

Lastly figure 9 should also include the winter and summer correlation plots, like that for spring and autumn.

Response: We added the the winter and summer correlation plots in Figure 9.

S6/ Section 3.4 and 3.5: NH3 and CO industrial emissions.

Industrial ammonia:

Throughout the manuscript there is a lack of detail and discussion on possible industrial (point source) emissions of NH3 (such as in Wang, et al, 2015). In section 3.4 analysis only concerns NH3 from vehicle sources. Section 3.5 talks about agricultural emissions (transported into the Hefei region) with a passing mention of "urban anthropogenic emissions" (L388). The only specific mention of point sources is in figure 1. Two large NH3 point sources are identified based on Table A1 of Clarisse et al. 2019.

Clarisse, L., Van Damme, M.,Clerbaux, C., and Coheur, P.-F.: Tracking down global NH3 point sources with wind-adjusted superresolution, Atmos. Meas. Tech., 12, 5457–5473, https://doi.org/10.5194/amt- 12-5457-2019, 2019.

Wang, S., Nan, J., Shi, C., Fu, Q., Gao, S., Wang, D., Cui, H., Saiz-Lopez, A., and Zhou, B.: Atmospheric ammonia and its impacts on regional air quality over the megacity of Shanghai, China, Sci. Rep.,5,15842, https://doi.org/10.1038/srep15842, 2015.

Industrial CO emissions:

L329 "To examine the contribution of traffic to NH3 columns, we analyze the relationship of NH3 columns with CO surface concentrations" and L342 "Atmospheric CO is regarded as a primary pollutant mainly emitted from vehicles in urban areas, and there is no significant biomass burning source around the Hefei site, thus the elevated NH3 columns are likely partly caused by urban emissions from vehicles.". Again, this is a lack of detail on CO industrial (point source) emissions and the effect this would have on any NH3-CO correlations. For example, there is a 700MW coal-fired power station (https://www.gem.wiki/Hefei-2_power_station) to the east of Hefei (~8km), the red flag in the image below, and two power stations in the Hefei region identified in Tan, et al. 2019.

[Figure]

Without detailed analysis that includes probable/possible industrial sources of NH3 and CO, conclusions drawn in sections 3.4 and 3.5 are highly speculative and qualitative (using vague phrases such as "likely partly" (L344)).

Tan, W., Zhao, S., Liu, C., Chan, K., L., Xie, Z., Zhu,Y., Su, W., Zhang, C., Liu, H., Xing, C., and Liu, J., Estimation of winter time NOx emissions in Hefei, a typical inland city of China, using mobile MAXDOAS observations, Atmos. Environ., 200, 228-242, https://doi.org/10.1016/j.atmosenv.2018.12.009, 2019.

Response: We added the discussion of possible industrial (point source) emissions of $NH_3$ and CO, including two power plants in Hefei, and two large $NH_3$ point sources identified by Clarisse et al. (2019), explaining the effect of industrial (point source) emissions on any NH3-CO correlation in Section 3.3 "Identification of emission sources of NH3".

S7/ Traffic pollutant correlations.

L336: "The results show that there exists no correlation between NH3 columns and concentrations of NO2, SO2, PM10 and the value of AQI over the four seasons (not shown)." Mathematically, there is always a correlation coefficient, if the correlation is less than 0.1, maybe better to state correlation is negligible. Even better, would be to list the correlation coefficient value after each species, i.e. NO2 (XX), SO2 (XX) etc…

It is interesting to read that there is no (negligible) correlation with NO2. NOX (NO2 + NO) is a primary pollutant from vehicle emissions (Zhou et al., 2014, Phillips et al, 2019). What does a NH3-CO moderate correlation with a negligible NH3-NO2 correlation indicate? Could the authors please comment on the reported NH3-NO2 correlation.

Zhou, Rui, et al. "Study on the traffic air pollution inside and outside a road tunnel in Shanghai, China." PloS one 9.11 (2014).

Phillips, F. A., et al.: Vehicle ammonia emissions measured in an urban environment in Sydney, Australia, using open path Fourier Transform Infra-Red Spectroscopy. Atmosphere, 10, 208, https://doi.org/10.3390/atmos10040208, 645 2019.

Response: We discuss the correlation of $NH_3$ with CO and $NO_2$ to find whether traffic

contributes to NH$_3$ emissions, so it is not important to discuss the relationships of NH$_3$ with SO$_2$, PM$_{10}$ and AQI. Therefore we removed the description of the correlation between NH3 columns and concentrations of SO$_2$, PM$_{10}$ and the AQI over the four seasons. We added the specific correlation coefficient between NH$_3$ and NO$_2$, and the description about what the NH3-CO moderate correlation with a negligible NH$_3$-NO$_2$ correlation indicates, in Section 3.3 "Identification of emission sources of NH$_3$".

There is also no quantitative measurand of traffic volumes. If NH3 emissions were primarily (or even partly) from vehicle emissions, then more in-depth quantitative analysis is required.

Response: Sorry we have not monitored the traffic volumes during the measurement, so we didn't use the value to quantitatively analyze the vehicle emissions.

S8/ NH3 and PM2.5.

L345: "Meanwhile, NH3 columns show weak correlation (R=0.47) with PM2.5 concentrations (Fig.7(b)), meaning that NH3 contributed to the formation of fine particulates significantly in summer.". This qualitative statement needs expanding. More in-depth analysis of the Hefei dataset is needed. A more quantitative investigation is required into PM2.5 and NH3 correlation and/or causation. It is well known NH3 contributes to PM2.5.

Response: Sorry the concentrations of ammonium( $NH_4^+$ ) and sulfate-nitrate-ammonium(SNA) aerosols near our Hefei site are not available, and the purpose of the section is to find the possible emission sources of NH3, so we removed the description about the relationship of NH3 and PM2.5 in the paper.

S9/ Column to point correlations

The NH$_3$ measurements are partial columns extending from ~0-4km in altitude. The meteorological and air quality datasets are from point source ground level measurements, located a distance from the NH3 measurements. These two datasets will have quite different footprints. The in-situ ground based measurements will be have a more localised footprint, whereas the column measurements will have a larger regional footprint. Could the authors comment on this fact and how it affects correlation interpretation.

Response: We added the comment on how the column with point correlations affects the correlation coefficient in section 3.3 "Identification of emission sources of NH$_3$".

S10/ Agricultural NH$_3$ emissions

L362 – 367: "In Hefei, the cropland in spring is characterized by the growing season of wheat and early rice, and synthetic fertilizer is applied during this period. The fertilizer application also tends to occur in autumn, when winter wheat is sowed, and a second rice crop is growing. The agricultural practice may explain the correlation between NH3 columns and temperature over Hefei over these two seasons, which suggests that agriculture was the main source of ammonia in spring and autumn."

This a good starting hypothesis to investigate, but it is left as a conjecture. Too vague language is used, e.g. "May explain" and "suggests". These are good starting points, but quantitative substantial evidence is required. This is lacking. Seasonal (hence temperature) correlations is not a strong causal link without additional collaborating evidence. The authors need to add such evidence to prove the statement in the manuscript.

Response: We added some descriptions from other publications to support our conclusions. However, quantification of $NH_3$ emissions from different sources in four seasons in Hefei is very difficult for us, as the High-resolution inventory of ammonia emissions in 2017 and 2018 in Hefei is not available, so we have no further quantitative data to prove the emission sources in four seasons.

S11/ Windrose

In such wind rose polar plots the wind strength is usually plotted as the radial component and the NH3 abundance as coloured points. I would strongly recommend replotting with wind speed as the radial axis of the polar plot and NH3 concentrations as coloured dots. Looking at the current plots, the greatest wind speed is less than 4.5ms-1. This is classified as a gentle breeze using the Beaufort scale. This indicates that the majority of measured NH3 is from local sources (assuming hourly- low daily NH3 lifetimes). The highest concentrations of NH3 is measured during easterly winds, the direction of the main Hefei urban area and the lowest concentrations from the west, a less populated and dense area. At L368 the authors comment on the wind rose data, but do not use it to help attribute the urban emission sources (traffic vs industrial). High NH3 concentration points are in the NW quadrant, the direction of an NH3 point source shown in figure 1. There is no commentary on this fact.

Response: We replotted Figure 10, with wind speed as the radial axis of the polar plot and $NH_3$ concentrations as colored dots. And we added comments on the wind rose data, using it to help attribute the urban emission sources, in section 3.3 "Identification of emission sources of $NH_3$".

S12/ NH3 lifetime.

There is only passing mention of NH3 lifetime (L55). Is there an expected difference in NH3 lifetime depending on emission source (i.e. rural vs industrial) and would this influence NH3 source attribution?

Response: $NH_3$ lifetime depends on other pollutants (reaction with ammonia), meteorological variables (water vapor, clouds and temperature), winds (atmospheric mixing and recirculation) and the $NH_3$ column itself. We read some publications, but didn't find there is difference in $NH_3$ lifetime depending on emission source (i.e. rural vs industrial).

S13/ Back-trajectory calculations using HYSPLIT.

At L380 the authors introduce a new method to help assess NH3 source emissions. Back trajectory modelling is introduced, but I cannot understand why the bulk of the findings are relegated to the supplement. I recommend that the supplement content and the main

manuscript from L380 -390 be concatenated into a new section, e.g. section 3.6.
In addition, a much more detailed explanation of the model set up (meteorological fields, resolution, cluster analysis details, etc) and analysis (source attribution) needs to be undertaken. Example: the PSCF's in figure S2 are hard to see in detail and require further inquiry. The interpretation of the results at L386-390, and the supplement, need more rigorous assessment in a scale more regional to Hefei (due to low wind speeds). There is no mention of carbon monoxide transport either, and the effect this would have on NH3-CO correlations and source attribution.

Response: We moved the supplement content to a new section 3.4. We added explanation of the model set up and and analysis in a scale more regional to Hefei. We also plotted the PSCF of CO concentrations over Hefei from 2017 to 2018 in four seasons. But it seems difficult to use it to explain the NH3-CO correlations and source attribution.

[Figure]

(a)                                    (b)

(c)                                    (d)

S14/ Concluding statements in section 4.
This section requires major revision based upon the issues raised in this review. Currently there is not enough collaborating evidence in the manuscript to explicitly state with certainty the origin and abundance of NH3 emissions measured at Hefei.
Response: We revised the concluding statements in section 4 based on the revision above.

**Minor/Technical:**
M1/ L30: "emission sources and potential sources". What is the difference between

emission and potential sources? Do you mean known emission sources and unknown emission sources? Can you please make this clearer and correct through-out the manuscript?

Response: We corrected the statement throughout the manuscript.

M2/ L34: "The results demonstrate the IASI data are in broad agreement with our FTIR data.". I think it best to rephrase as "The results demonstrate that the IASI and FTIR data, over Hefei, are in broad agreement.".

Response: We corrected this sentence.

M3/ L64: "This is mainly caused by lack of representative measurements of atmospheric NH3.". Could a reference to this statement be added?

Response: We added a reference here.

M4/ NH4+: There is no mention of NH4+ with respect to NH3 partition equilibrium in NHx, and how this will affect the measured NH3 abundances. Could the authors please comment on this aspect.

Lachatre, et al, 2019 would be a good starting point.

Response: We added the comment about NH4+ with respect to NH3 partition equilibrium in 3.1 "Time series and seasonal trend of NH3".

M5/ L68: There are ~12 references to a single statement. Can this be cut down to only the essential references needed. Passive sampling is not the focus of this manuscript.

Response: We removed 7 references and kept 5 essential references needed.

M6/ L71: Pedantic but…could 'Sticky' be replaced with a more scientific descriptive term, or place in quotation i.e. 'sticky'.

Response: We replaced 'Sticky' with 'and easy to adsorb onto surfaces'.

M7/ L78: Replace 'ground-based' with 'ground level'

Response: We replaced 'ground-based' with 'ground level'.

M8/ L120: "Furthermore, ground based observations of NH3 have been sparse throughout China (Liu et al., 2011; Xu et al., 2015)." Ground based or ground level in situ?

Rasponse: We added 'ground level in situ' in the sentence.

M9/ L132: Replace "reduces" with "reductions of"

Response: We replaced "reduces" with "reductions of"

M10/ L145: I think "Materials" is the incorrect word, maybe "Measurements" would be better.

Response: We replaced "Materials" with "Measurements".

M11/ L164: "liquid-nitrogen-cooled MCT/InSb detector". Just to clarify, is this a single detector or dual HgCT and InSb detectors?
Response: We replaced "liquid-nitrogen-cooled MCT/InSb detector" with "liquid-nitrogen-cooled MCT and InSb detector".

M12/ L165: Replace 'suit' with 'suite'
Response: We replaced 'suit' with 'suite'.

M13/ L171: "At the same time, the indoor pressure, temperature and relative humidity are logge continuously.". Are these important parameters for the FTIR measurements?
Response: We removed this sentence.

M14/ L173: "similar to the NH3 retrieval strategies in Dammers et al. (2015)". What are the differences? Are the Microwindows the same?

Response: We replaced "similar to the NH3 retrieval strategies in Dammers et al.

(2015)" with "same as the $NH_3$ retrieval microwindows adopted by Réunion and Jungfraujoch sites in Dammers et al. (2015)".

M15/ L181: "A priori profiles of NH3 and interfering gases are taken from the Whole Atmosphere Community Climate Model (WACCM, v.6_120_99) in combination with initial measurement values." There is no NH3 modelling output from WACCMv6. Could the authors please correctly reference the NH3 apriori source used. Is a static apriori used? Or does it change throughout the retrieval time series?
Response: We correctly referenced the NH3 a priori source used. It is a static a priori profile used.

M16/ L181: "…in combination with initial measurement values.". Sorry, I do not understand this. Can you please explain what initial measurements are, and how they are combined with (so-called) WACCMv6 profiles to obtain an apriori?

Response: We replaced "…in combination with initial measurement values." with "A

priori profile of NH3 is taken from the one adopted in Dammers et al. (2015), which is based on the average Bremen NH3 concentration, and then adjusted to match the local surface concentrations measured in our site."

M17/ L182: "WACCM, v.6_120_99". I think this can be referenced better. '_120_99' is not part of the WACCM simulation identify code. Maybe better to reference it this way: (WACCMv6, Garcia et al. 2007).
Garcia, R. R.,Marsh, D. R., Kinnison, D. E., Boville, B. A., and Sassi, F.: Simulation of secular trends in the middle atmosphere, 1950-2003, 20 J. Geophys. Res., 112, D09301, doi:10.1029/2006JD007485, 2007.
Response: We referenced the WACCM as   (WACCMv6, Garcia et al. 2007).

M18/ L183: "for ammonia was constructed to be diagonal, with standard deviations of 100% for all layers." What was the basis for assuming an uncertainty of 100% and is there any inter-layer covariance/correlation constraint?

Response: We added the basis for assuming an uncertainty of 100% and there is a Gaussian interlayer correlation with HWHM of 4 km after check the input file of retrieval.

M19/ I would also encourage the authors to include another sentence (or two) after the description of the apriori profile and uncertainties describing other state vector retrieval parameters such as phase fitting, instrument line shape, continuum fit, zero level offset etc.

Response: We added the description about the other state vector retrieval parameters, such as phase, instrument line shape, continuum fit, zero level offset.

M20/ Also…could be authors also state the SNR (or Se) of the measurements, and either if the SNR is held static or varies on a spectrum by spectrum basis.

Response: We stated the SNR of the measurements, and we set the SNR to be 200 for each spectrum.

M21/ How is the H2O apriori set? For instance, is it from NCEP daily profiles, or from WACCM, or from a previous H2O retrieval?

Response: The water profiles are derived from NCEP reanalysis daily mean specific humidity profiles.

M22/ L190: "about 0.498 % and 0.505%", could these values be rounded to the correct number of significant figures.

Response: We round them to the correct number.

M23/ Fig 2 and L188: "The measured spectrum is shown in blue, the fitted spectrum in red and the residual in black". There are two blue lines, 'observed' and H2O. Either state as 'dark blue' or switch colour.

Response: We replotted the Figure 2.

M24/ I recommend that section 3.1.2 lines L238 to 246 be moved to the end of section 2.2. This is because it details retrieval information content and logically comes after spectral fitting.

Response: We moved the contents about lines L238 to 246 to the end of section 2.2.

M25/ L239: "It is evident that the retrieval is most sensitive to the troposphere, where the concentration of ammonia peaks." Add in 'lower': "It is evident that the retrieval is most sensitive in the lower troposphere, where the concentration of ammonia peaks"

Response: We corrected the sentence.

M26/ L243: "The degrees of freedom for signal (DOFS) value is 1.10 given by this measurement". Can the authors indicate at what height, from the surface up, does the DOFS reach 1.0, essentially showing the partial column range in which there is a signal?

Response: DOFS reach 1.0 at about 3.2 km from the surface up. We stated it.

M27/ Fig 4. The individual layer averaging kernels are all the same colour, very confusing. Maybe only plot for layers 0-10km. There are also no units for the VMR averaging kernels (I assume VMR/VMR)? Also, for the total column averaging kernel, what are the units?

Response: We replotted Fig 4, now is Fig 3, according to the suggestions.

M28/ Section 3.1.1 could be transferred to be section 2.3. (IASI becomes section 2.4). It makes more sense that the entire retrieval process from retrieval set up to retrieval output then error analysis is all together in one section.

Response: We transferred Section 3.1.1 to be section 2.3, and "IASI data" becomes section 2.4.

M29/ L215: "The error calculation is based on attributing uncertainties to all parameters used in the profile retrieval". Maybe better stated as "The error calculation is based on attributing uncertainties to all parameters in the retrieval state vector".

Response: We corrected this sentence.

M30/ L221: "Table 1 lists the uncertainties of the parameters assumed in the retrievals." Maybe better stated as: "Table 1 lists the uncertainties of the retrieval state vector parameters."

Response: We corrected this sentence.

M31/ L222: "The total errors are about 11.42 % based on the combination of random and systematic errors". Is error analysis applied to each individual retrieval? If so, the mean total error and 1-sigma standard deviation of the time series could be given. This would be a better indication of the overall timeseries uncertainty than a single retrieval.

Response: We gave the the mean total error and 1-sigma standard deviation of the time series here.

M32/ Table 1:
Could the authors please state where or how the temperature profile uncertainties were set/calculated. Did the authors calculate the uncertainty associated with inferring species spectral uncertainties? i.e. the line intensity (and pressure and temperature broadening) uncertainties in $CO_2$, $H_2O$ etc…
Could 'NH3' be put in front of Line Intensity, and P, T broadening (broadening, not broading). Could the authors also reference the origin of the 10% spectral uncertainties.

Response: We redid error calculations considering the line intensity (and pressure and temperature broadening) uncertainties in interfering gases, such as $CO_2$, $H_2O$, and $O_3$. We changed the 10% of spectral uncertainties of $NH_3$ to 20%, according to the value in HITRAN 2012 database.

M33/ Section 3.1.2 and Fig 3: With Section 3.1.1 moved to section 2, along with the second half of section 3.1.2, this leaves a single paragraph (L230-237) in section 3.1.2. This paragraph could be moved into section 3.2. There is very little vertical information due to the low DOFs, basically profile scaling.
Response: We moved the second half of section 3.1.2 to section 3.2.

Figure 3 offers little information due to the low DOFs. The variation within a season is not presented so it is hard to conclude if the seasonal differences in profile are significant. Maybe reconstruct figure 3 to have 4 plots (2 by 2), one for each season and for each season include the spread (1-sigma SD?) at each level.
Response: We replotted Fig. 3 (now Fig. 4) to have 4 plots (2 by 2) and include the spread (1-sigma SD) at each level.

M34/ L235: "The seasonal averaged surface level of NH3 decreased from 10.82 ppb in summer to 2.92 ppb in winter during 2017 and 2018, and the corresponding values are about 5.48 and 6.04 ppb in spring and autumn, respectively.". It would be advantageous if for each season the mean and spread (1-sigma SD) was given.
Response: We gave the the mean and spread (1-sigma SD) of surface concentration for each season.

M35/ L250: "Many spectra ranging from 700 to 1350 cm-1 are saturated in summer (due to high humidity), causing the retrieved NH3 data to be sparsely sampled relative to those in other seasons". MW1 (~929.6-931.2cm-1) (fig 2a) shows small H2O absorption features, as opposed to MW2. In summertime, could plausible retrievals be performed with only using MW1 without altering the DOFS too much? This may help increase the number of observations.
Response: We tested this method only using MW1, but this changed the results slightly.

M36/ L276: "It follows therefore that seasonal variation of NH3 columns in the Hefei area accords with that in other areas in China, with the main emission source being agriculture." This sentence is not needed as a near repeat of the statement at L263.
Response: We removed this sentence.

M37/ L287: Maybe rewritten as "Here we present a comparison of the ground-based NH3 column measurements with the IASI satellite measurements".
Response: We rewritten this sentence.

M38/ L290. How was the co-location criteria of 0.5 degrees and 90 minutes decided

upon?

Response: We gave the reason why set the co-location criteria.

M39/ L303: "The Relative differences larger than 100% were considered as outliers from the data". On what basis, and why the value of 100%? Was the ground based or satellite measurements anomalous?

Response: We found that the satellite measurements with large retrieval errors when the relative differences is larger than 100%.

M40/ L305: "with standard deviation of 44.44% and 41.00%, respectively.". Maybe 44% and 41%.

Response: We corrected the two values. The IASI team provided the new version of IASI data. We compared the IASI data of new version with our FTIR data, so the comparison results are slightly different with the previous ones.

M41/ L308: "Additionally, the distributions of the relative difference of the two dataset show that, the relative bias mainly range from -60% to 80% for IASI A data, from -60% to 60% for IASI B data, and the bias from -20% to 0% as well as -40% to -20% has the highest frequency in both respective bins (Figure 6 (c) and (d))." Bias refers to the overall difference between the two datasets, so I think it is better worded as: "Additionally, the distributions of the relative difference of the two datasets show that the relative differences of a measurement pair, mainly range from -60% to 80% for IASI A data, from -60% to 60% for IASI B data as seen in figure 6 (c) and (d)."

Response: We corrected this sentence.

M42/ L313L First time the acronym NDACC is used. For the first instance can the full title please be used. Also maybe better stated as: "nine NDACC FTIR stations".

Response: We gave the full title of NDACC and corrected this statement.

M43/ L317: "The correlation coefficient (R) between the CrIS NH3 total columns and FTIR data was 0.77, and the relative difference is 0-5 % with a standard deviation of 25–50 % for comparison of high levels of NH3." Sorry I do not understand the '0-5%' statement, can you please explain it more clearly, how can you have multiple mean differences?

Response: We explained this sentence more clearly.

M44/ L320: "results from other NDACC site data". Is Hefei a NDACC affiliated FTIR site? If not, then the current wording can be read that it is. Maybe replace with "comparison results in the aforementioned literature"

Response: We corrected this sentence.

M45/ L330: "The Dongpu Reservoir air quality monitoring station (31.91°N, 117.16°E) is very close to our site, part of a National Ambient Air Quality Monitoring Network, which monitors and routinely publishes the concentrations of main gaseous

pollutants, including CO, NO2, PM2.5, PM10, SO2, O3 and Air Quality Index (AQI) etc". Could the distance from the air quality station to the FTIR site please be stated and a reference added pertaining to the National Ambient Air Quality Monitoring Network, (esp. for the Hefei site if possible). Could the Dongpu site please be marked in figure 1b.

Response: We stated the distance from the air quality station, and added a reference about the National Ambient Air Quality Monitoring Network. We also marked the the Dongpu site in Fig. 1b.

M46/ L376: "Overall, the results indicate that air temperature, wind direction and wind speed are the main factors that influence gaseous NH3 concentrations in Hefei.". What about source emissions?

Response: We corrected this sentence as "Overall, the results indicate that air

temperature, wind direction and wind speed in addition to source emissions are the main factors that influence gaseous $NH_3$ concentrations in Hefei.".

M47/ L393: the adjective 'great' can be omitted.
Response: We removed this word.

M48/ L415: replace "meteorological parameters" with "meteorological conditions".
Response: We replaced this word in this sentence.

M49/ L432: "This is the first time that ground-based FTIR remote sensing of NH3 columns and comparison with satellite data are reported in China". A valid and important finding and should be reported at the start of the conclusion section.
Response: We moved this sentence to the end of the first paragraph in section 4 "Conclusions".

M50/ L405: "To validate the satellite observations…". In this case since the satellite measurements have been compared against other measurements favourably, the satellite measurements are used to confirm/validate the Hefei site ground base measurements.
Response: We rewrote the sentence "Since the IASI satellite measurements have been compared and validated by other measurements, the IASI satellite data were used to confirm the ground-based measurements at Hefei site.".

M51/ Fig 1: Could longitude and latitude units please be put on both axis. The area displayed could also be reduced to focus on the eastern and middle region of China. In map B, could the Dongpu AQ station also be identified. Could the location identifier 'dots' also be bigger in both maps. Could a distance scale also be put on the map? Caption: "The regional distributions of NH3 columns (molec cm-2) from 2008-2018 IASI-A and 2013-2018 IASI-B morning overpasses of ANNI-NH3-v3R data". Are these annual mean columns? Good idea to specify the temporal period.

Response: We replotted Fig 1 following the suggestions.

M52/ Fig 2a and b: the title consists of RMS. If possible, add in the DOFS and CHI^2 fit variables as well for completeness.
Response: We added DOFS and CHI^2 in the caption of Fig 2.

M53/ Fig 5: Could 1-sigma standard deviation spread/error bars be put on the daily averages.
Response: We replotted Fig 5, with 1-sigma standard deviation.

M54/ Fig 6: A lot of white space in plots A &B, could the X-axis range be shortened, MAX X ~= 8e16.
Response: We didn't shorten the X-axis range, as we want to plot the the line y=x, which need the X-axis range is same with the Y-axis one.

M55/ Fig 7: Could the uncertainty bars be added for each point, like that in fig 6 a and b. Caption: "Scatter plot of NH3 columns (molec cm-2) with CO (mg m-3, a) and PM2.5 (μg m-3, b) concentrations measured at the Dongpu Reservoir air quality monitoring site in summer." Are these daily averages? Please add in temporal period.
Response: We replotted Fig. 7, with 1-sigma standard deviation for CO.

M56/ Fig 9: Caption: "Scatter plot of NH3 column (molec cm-2) with air temperature (°C) in spring (a) and autumn (b)." Again, please add temporal period (hourly, daily, monthly…).
Response: We added "individual NH$_3$ column" here.

M57/ Fig S1 and S2, it is hard to ascertain information from both plots. As a first step, could the map lat/lon boundaries be reduced to concentrate on areas of interest better. I.e. zoomed in a bit.
Response: We replotted Fig S1 and S2 (now Fig 11 and 12).

M58/
Franco, B. Clarisse, L., Stavrakou, T., Müller, J.-F, Van Damme, M. ,Whitburn, S., Hadji-Lazaro, J. , Hurtmans, D., Taraborrelli, D. , Clerbaux,C. , Coheur,P.-F.: A General Framework for Global Retrievals of Trace Gases From IASI: Application to Methanol, Formic Acid, and PAN, J. Geophys. Res.-Atmos.,
123, 963-984, https://doi.org/10.1029/2018JD029633, 2018.
Is this reference relevant?
Response: We kept this reference as we think it is relevant to the retrieval methods about IASI data.

---

## Author Comment (AC2) · 8 Jul 2020

**Response to comments #2**

We appreciate your constructive and positive comments. The comments and proposed corrections have been taken into account and helped to improve the paper. Each comment has been addressed as follows. There is an extensive discussion among the authors regarding how to revise the content. So the response is delayed, and we are sorry for this.

**General comments**

This paper presents ground based FTIR measurements at the Hefei site including error analysis of the NH3 retrievals, vertical distribution, time series and seasonal trend analyses. More additional works such as comparisons of IASI data, relationship with surface CO, temperature, wind speed and direction, and back trajectories analysis are made in this paper. I believe this paper is suitable for publication to Atmos. Meas. Tech. after considering comments as below. As for dividing into two papers by anther refree, if authors can prepare more analyses and discussions for AMT and another paper, I think it would be better. If not, One paper of AMT looks not bad.

Response: We didn't divide the paper into two papers after discussion among the authors

**Specific comments**

line 2

I feel "measured" is not fitted because of "measured from observations". Retrieved, derived, obtained etc. would be better.

Response: We replaced "measured" with "retrieved".

Line 32 and 402

If possible, could you provide error (one standard deviation) for 22.14 % yr-1 annual increase rate?

Response: We replaced "annual increase rate" with "increase of the annual mean", as another reviewer think it is premature to report the annual trend. So we gave the annual mean and one standard deviation for the two years.

Line 38-39

"Further, high correlation of NH3 columns with air temperature is obvious from their diurnal variation during the observation period."

"In addition, the clear correlation between NH3 columns and air temperature in spring and autumn over Hefei, suggests that agriculture was indeed the main source of ammonia in spring and autumn."

I think a correlation coefficient NH3 columns with air temperature should be provided, since NH3 columns with CO concentration is described as R=0.77.

Response: We calculated the correlation coefficient between NH$_3$ columns with air

temperature from their diurnal variation, and the correlation coefficient is 0.39, showing a weak correlation between $NH_3$ and temperature from their diurnal variation. The $NH_3$ data measured after 14:00 PM local time are sparse as we collect the MIR and NIR solar spectra alternately every day, so the data in some time period lacks of representation for diurnal variation analysis. Therefore we removed the discussion about the correlation of $NH_3$ columns with air temperature from their diurnal variation, including Figure 8, in section 3.3 "Identification of emission sources of $NH_3$".

Line 83-89
Authors had better add the GOSAT retrieval from TANSO-FTS TIR spectra as recent results.
Citation: Someya, Y., Imasu, R., Shiomi, K., and Saitoh, N.: Atmospheric ammonia retrieval from
the TANSO-FTS/GOSAT thermal infrared sounder, Atmos. Meas. Tech., 13, 309–321, https://doi.org/10.5194/amt-13-309-2020, 2020.
Response: We added the GOSAT retrieval from TANSO-FTS TIR spectra as recent results, and included the citation.

Line 99-102
"More recently, FTIR measurements have been shown to also provide total column and vertical profiles of ammonia at a high temporal resolution, and are now also used for validation of satellite NH3 observations (Dammers, et al., 2015; Dammers, et al., 2016; Dammers, et al., 2017b)."
I recommend to adding a name of the satellite, that is, IASI.
Response: We added the name of the satellites, i. e. "IASI and CrIS".

Line 226
Some of working in same field can understand "phase", but for wider readers, a little explanation might be necessary.
Response: We added the description of "phase", including two citations.

Line 250
"Many spectra ranging from 700 to 1350 cm-1 are saturated in summer (due to high humidity), causing the retrieved NH3 data to be sparsely sampled relative to those in other seasons."
Are there else any better spectral windows for retrieval of NH3 in summer season?
Response: We tested a few spectral windows for retrieval of $NH_3$ in summer season, and chose the two microwindows, according to the fitting RMS, DOFS and other output parameters.

Line 281
"The annual increasing rate of ammonia columns in Hefei estimated by our two-year FTIR measurements (22.14 % yr-1) is much larger than the reported value by satellite observations over China. This is likely due to the different sampling years. The

increasing trend of NH3 in Hefei is likely caused by either an increased fertilizer use, or increasing air temperature, or decreased sulfur emissions due to strict SO2 control measures."

Is it possible to verify the annual increasing rate of ammonia columns over Hefei using other satellite or model data?

Response: We used the IASI data from 2016 to 2018 over Hefei to calculate the annual increase of ammonia columns. The increase of NH$_3$ is 19.66% and 8.92% for the annual mean from 2016 to 2017, and from 2017 to 2018, respectively, for IASI-A data. It is 5.97% and 5.13% for the annual mean from 2016 to 2017, and from 2017 to 2018, respectively, for IASI-B data. The increase values are very different for IASI themselves. I don't know the reason. So I didn't use other satellite or model data to verify the annual increasing rate of ammonia columns over Hefei.

Line 291

"We remove the data with negative IASI-NH3 columns due to large retrieval error."

Negative values for the IASI-NH3 columns are not physically meaning. I think large retrieval error is not fundamental reason.

Response: We corrected this sentence.

Line 303-311

There would be different results comparing of IASI A and IASI B data with the Hefei FTIR data, but they are within one standard deviation. Could you describe this reason by citing the literatures or technical reports from the IASI team? If impossible, a description that the difference is within one standard deviation would be there.

Response: The IASI team provided the new version of IASI data. We compared the IASI data of new version with FTIR data, so the comparison results are slightly different with the previous ones. We added the description that the difference is within one standard deviation.

Line 319-321

"So the relative differences between IASI total columns and our FTIR data and standard deviations of the differences are within the range of comparison results from other NDACC site data, and the correlation coefficients are comparable to that of other comparison results."

In table 3 of Dammers et al. (2017b) paper, the mean relative difference (MRD) at Wollongong site is only positive (6.0 ± (74.3)%) and other sites are negative. If readers know this, readers may get confused. More detail discussions and descriptions are necessary.

Response: We added discussions about the relative difference.

Line 338

"However, NH3 columns show high correlation with CO concentrations in summer, as displayed in Figure 7(a)."

How is other seasons ?

Response: We added "NH$_3$ columns show negligible correlation with CO concentrations in other seasons."

Line 345

"Meanwhile, NH3 columns show weak correlation (R=0.47) with PM2.5 concentrations (Fig. 7(b)), meaning that NH3 contributed to the formation of fine particulates significantly in summer."
If so, a correlation with PM2.5 concentrations in other seasons might be higher than that in summer. Did authors check them?
Response: We calculated the correlation of NH$_3$ columns with PM$_{2.5}$ concentrations, but there are negligible correlations in other seasons. Sorry the concentrations of ammonium($NH_4^+$) and sulfate-nitrate- ammonium(SNA) aerosols near our Hefei site are not available, it is difficult to assess the contribution of NH$_3$ to PM$_{2.5}$. Also, the purpose of the section is to find the possible emission sources of NH$_3$, so we removed the description about the relationship of NH3 and PM2.5 in the paper.

Line 352

"High correlation of NH3 columns with air temperature is obvious from their diurnal variation during the observation period, as seen in Figure 8. Our measurements are performed generally from 9:00 to 16:00 local time. The whole data are averaged per hour during the two years." Considering discussions that follow, a plot in Figure 8 prepared for whole data (I understand all seasons) should be prepared for each seasons. Could authors explain a reason that NH3 columns decreased from 11:30 to 13:30 in figure 8? If plots are prepared for each season, decreasing in spring and autumn might be appeared.
Response: We ever plotted Fig.8 for each seasons, but the variations of temperature and NH$_3$ columns seem unreasonable. Also, We calculated the correlation coefficient between NH$_3$ columns with air temperature from their diurnal variation, and the correlation coefficient is 0.39, showing a weak correlation between NH$_3$ and temperature from their diurnal variation. The NH$_3$ data measured after 14:00 PM local time are sparse as we collect the MIR and NIR solar spectra alternately every day, so the data in some time period lacks of representation for diurnal variation analysis. Therefore we removed the discussion about the correlation of NH$_3$ columns with air temperature from their diurnal variation, including Figure 8, in section 3.3 "Identification of emission sources of NH3".

**Technical corrections**
line 28-29
I fell there is a duplication. One idea is to remove "a measurement site in".

Response: We removed "a measurement site in".

Line 32

measurement-> measurements
Response: We corrected this word.

Line 35
Analyze -> analyzed?
Response: We corrected this word.

Line 143
Remove "retrieved"
Response: We removed this word.

Line 169
"recorded"-> "has been recording"
Response: We corrected this word.

Line 179
"vertical profile"->" vertical profiles"
Response: We corrected this phrase.

Line 204
"general"-> "generally"
Response: We corrected this word.

Line 235
"The seasonal averaged surface level of NH3 decreased from 10.82 ppb in summer to 2.92 ppb in winter during 2017 and 2018, and the corresponding values are about 5.48 and 6.04 ppb in spring and autumn, respectively."
Line 261
"The annual mean NH3 column is $1.31\times10^{16}$ and $1.60\times10^{16}$ molec cm-2, respectively, with an increase rate of about 22.14 %."
If readers can know errors or standard deviation, they might be good.
Response: We added the standard deviation for these values.

Line 243, 244, 249 and other
"retrievals at the Hefei site" or "retrievals in the Hefei site." A lot of inconsistency, "at the Hefei site" might to be good.
Response: We corrected this phrase throughout the paper.

Line 264
If authors use "practices" as a noun, I think "maybe" is adverb and there is no verb in this sentence.
Response: We corrected this sentence.

Line 269

"agriculture"-> agricultural area?
Response: We corrected this phrase.

Line 325
"tunnel studies"
Simple description for them is grateful.
Response: We added simple description for tunnel studies.

Line 330
"The Dongpu Reservoir air quality monitoring station (31.91°N, 117.16°E) is very close to our site, part of a National Ambient Air Quality Monitoring Network, which monitors and routinely publishes the concentrations of main gaseous pollutants, including CO, NO2, PM2.5, PM10, SO2, O3 and Air Quality Index (AQI) etc."
There is no citation for the data.
Response: We added a citation for the air quality data.

Line 434
"Future work" -> "Future works"
Response: We corrected this word.

Line 435
"to estimate regional"->"estimating regional" or "estimation of regional"
Response: We corrected this phrase.

Figure 1 caption
"The regional distributions of NH3 columns (molec cm-2) from 2008-2018 IASI-A and 2013-2018 IASI-B morning overpasses of ANNI-NH3-v3R data."
Are they averaged values or overlaid ? Clarification would be necessary.
Response: They are averaged values. We clarified this.

Figure 4 (a)
Higher than 40 km should be removed for a color bar for the altitude and replot would be necessary. If Authors can do them, readers may understand which colors are altitudes for VMR averaging kernels. But I don't know it is useful.
Response: We replotted Figure 4, removed the part higher than 20km.

Figure 7 (b)
"ug"->"micro g" Micro is small Greek letter.
Response: we removed Fig. 7(b), as we removed the description about the relationship of $NH_3$ column with $PM_{2.5}$ concentration.

Figure 10
Digits after the decimal point might be not necessary for NH3 column. Also digits after the first decimal point might be not necessary for wind speed.

I feel wind speed would for radial axes and NH3 columns for color bars would be better for better understanding relationship of NH3 columns to wind direction and speed. If authors did not try, please try.

Response: We replotted Fig. 10, with wind speed for radial axes and $NH_3$ columns for color bars.

Figure S1.

Back trajectories colored with black are very cloudy, if possible could authors color them for each cluster?

Response: Sorry it is difficult to color back trajectories for each cluster. We colored them using black dashed lines to replace black solid lines, to reduce cloudy.

Figures S1 and S2

What is light blue curved lines? There is no description for them.

Response: The light blue curved lines are rivers in China. Some panels showed them, while others didn't show. We replotted Figure S1 and S2 (Now Fig. 11 and Fig.12), making the light blue curved lines disappear for consistency.